# DIFFUSION STATE-GUIDED PROJECTED GRADIENT FOR INVERSE PROBLEMS

**Rayhan Zirvi**[*]**, Bahareh Tolooshams**[*]**, & Anima Anandkumar**
Computing and Mathematical Sciences
California Institute of Technology
`{rayhanzirvi,btoloosh,anima}@caltech.edu`
[*] Equal contribution

## ABSTRACT

Recent advancements in diffusion models have been effective in learning data priors for solving inverse problems. They leverage diffusion sampling steps for inducing a data prior while using a measurement guidance gradient at each step to impose data consistency. For general inverse problems, approximations are needed when an unconditionally trained diffusion model is used since the measurement likelihood is intractable, leading to inaccurate posterior sampling. In other words, due to their approximations, these methods fail to preserve the generation process on the data manifold defined by the diffusion prior, leading to artifacts in applications such as image restoration. To enhance the performance and robustness of diffusion models in solving inverse problems, we propose *Diffusion State-Guided Projected Gradient* (DiffStateGrad), which projects the measurement gradient onto a subspace that is a low-rank approximation of an intermediate state of the diffusion process. DiffStateGrad, as a module, can be added to a wide range of diffusion-based inverse solvers to improve the preservation of the diffusion process on the prior manifold and filter out artifact-inducing components. We highlight that DiffStateGrad improves the robustness of diffusion models in terms of the choice of measurement guidance step size and noise while improving the worst-case performance. Finally, we demonstrate that DiffStateGrad improves upon the state-of-the-art on linear and nonlinear image restoration inverse problems. Our code is available at `https://github.com/Anima-Lab/DiffStateGrad`.

## 1 INTRODUCTION

Inverse problems are ubiquitous in science and engineering, playing a crucial role in simulation-based scientific discovery and real-world applications (Groetsch & Groetsch, 1993). They arise in fields such as medical imaging, remote sensing, astrophysics, computational neuroscience, molecular dynamics simulations, systems biology, and generally solving partial differential equations (PDEs). Inverse problems aim to recover an unknown signal $\boldsymbol{x}^\star \in \mathbb{R}^n$ from noisy observations

$$\boldsymbol{y} = \mathcal{A}(\boldsymbol{x}^\star) + \boldsymbol{n} \in \mathbb{R}^m, \tag{1}$$

where $\mathcal{A}$ denotes the measurement operator, and $\boldsymbol{n}$ is the noise. Inverse problems are ill-posed, i.e., in the absence of a structure governing the underlying desired signal $\boldsymbol{x}$, many solutions can explain the measurements $\boldsymbol{y}$. In the Bayesian framework, this structure is translated into a *prior* $p(\boldsymbol{x})$, which can be combined with the *likelihood term* $p(\boldsymbol{y}|\boldsymbol{x})$ to define a *posterior distribution* $p(\boldsymbol{x}|\boldsymbol{y}) \propto p(\boldsymbol{y}|\boldsymbol{x})p(\boldsymbol{x})$. Hence, solving the inverse problem translates into performing a Maximum a Posteriori (MAP) estimation or drawing high-probability samples from the posterior (Stuart, 2010). Given the forward model $p(\boldsymbol{y}|\boldsymbol{x})$, the critical step is to choose the prior $p(\boldsymbol{x})$, which is often challenging; one needs domain knowledge to define a prior or a large amount of data to learn it.

Prior works consider sparse priors and provide a theoretical analysis of conditions for the unique recovery of data, a problem known as compressed sensing (Donoho, 2006; Candès et al., 2006). Sparse priors have shown usefulness in medical imaging (Lustig et al., 2007), computational neuroscience (Olshausen & Field, 1997), and engineering applications. This approach is categorized into model-based priors where a structure is assumed on the signal instead of being learned.

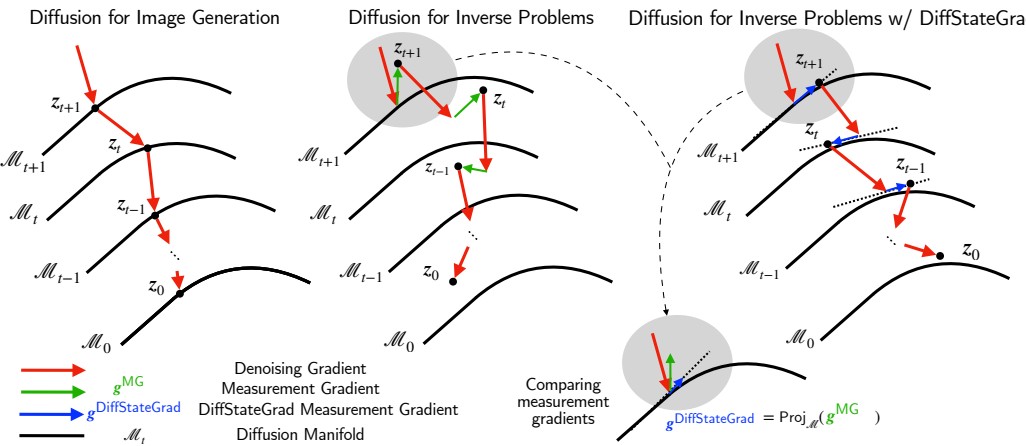

Figure 1: **High-level interpretation of Diffusion State-Guided Projected Gradient (DiffState-Grad).** DiffStateGrad projects the measurement gradient onto a subspace defined to capture statistics of the diffusion state at time $t$ on which the gradient guidance is applied. This helps the process stay closer to the data manifold during the diffusion process, resulting in better posterior sampling. Without such projection, the measurement gradient pushes the process off the data manifold. For when the measurement gradient guidance is applied to $z_{0|t}$, the projection is defined to capture the structure of the tangent space of the clean data manifold. The dotted straight line conceptually visualizes the subspace to which the measurement gradient is projected.

Recent literature goes beyond such model-based priors and leverages information from data. The latest works employ generative diffusion models (Song & Ermon, 2019; Kadkhodaie & Simoncelli, 2021), which implicitly learn the data prior $p(\boldsymbol{x})$ by learning a process that transforms noise into samples from a complex data distribution. For inverse problems, this reverse generation process is guided by the likelihood $p(\boldsymbol{y}|\boldsymbol{x})$, forming a denoiser posterior, to generate data-consistent samples. While diffusion models are state-of-the-arts, they still face challenges in solving inverse problems.

The main challenge arises from the fact that the denoiser posterior, specifically the likelihood component $p(\boldsymbol{y}|\boldsymbol{x})$, is intractable since the diffusion is trained unconditionally (Song et al., 2021). Prior work addresses this challenge by proposing various approximations or projections to the gradient related to the measurement likelihood $p(\boldsymbol{y}|\boldsymbol{x})$ to achieve likely solutions (Kawar et al., 2022); when these approximations are not valid, it results in inaccurate posterior sampling and the introduction of "artifacts" in the reconstructed data (Chung et al., 2023). *Latent* diffusion models (LDMs) (Rombach et al., 2022), due to the nonlinearity of the latent-to-pixel decoder, further exacerbate this challenge. Besides this approximation, the lack of robustness of diffusion models to the measurement gradient step size (Peng et al., 2024) and the measurement noise, and the lack of guarantees for worst-case performance limits their practical applications for inverse problems.

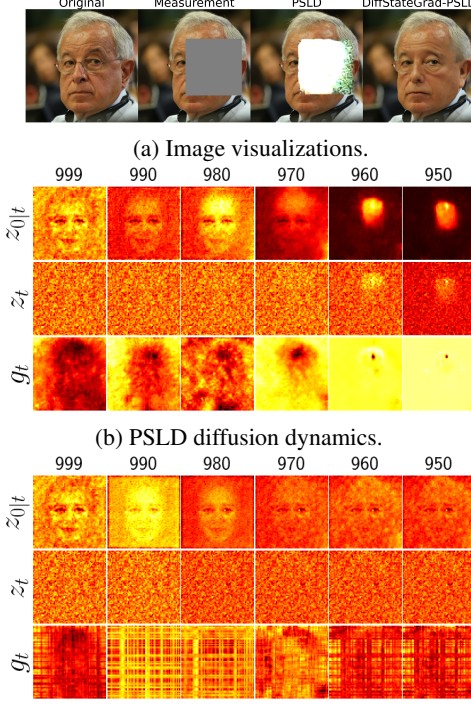

(a) Image visualizations.

(b) PSLD diffusion dynamics.

(c) DiffStateGrad-PSLD diffusion dynamics.

Figure 2: **Visualization of DiffStateGrad in removing artifacts.** The large MG step size pushes the process away from the manifold in PSLD, while DiffStateGrad-PSLD is unaffected. The title refers to the diffusion steps.

**Our contributions:** We propose a *Diffusion State-Guided Projected Gradient* (DiffStateGrad) to address the challenge of staying on the data manifold in solving inverse problems. We focus on gradient-based measurement guidance approaches that use the measurement as guidance to move

the intermediate diffusion state $x_t$ toward high-probability regions of the posterior. DiffStateGrad projects the measurement guidance gradient onto a low-rank subspace, capturing the data statistics of the learned prior (Figure 1). We visualize how the diffusion process is pushed off the manifold when the measurement step size is relatively large in a diffusion model and how the incorporation of DiffStateGrad alleviates this challenge (Figure 2). We define a projection step to preserve the measurement gradient on the tangent space of the state manifold. We achieve this projection by performing singular value decomposition (SVD) on the diffusion state of an image to which guidance is applied and use the $r$ highest contributing singular vectors as a choice of our projection matrix; by projecting the measurement gradient onto our proposed subspace, we aim to remove the directions orthogonal to the local manifold structure.

- We show that the crucial factor is the choice of the subspace, not the low-rank nature of the subspace projection. We find that our DiffStateGrad enhances performance. Our projection defines the subspace based on the structure of the state to which the measurement guidance is applied. This is in contrast to random subspace projections or low-rank approximations (Table 1).
- We theoretically prove how DiffStateGrad helps the samples remain on or close to the manifold, hence improving reconstruction quality (Proposition 1).
- We demonstrate that DiffStateGrad increases the robustness of diffusion models to the measurement guidance gradient step size (Figure 5, Table 5) and the measurement noise (Figure 6). For example, for a large step size, DiffStateGrad drastically improves the LPIPS of PSLD (Rout et al., 2023) from $0.463$ to $0.165$ on random inpainting. For large measurement noise, DiffStateGrad improves the SSIM of DAPS (Zhang et al., 2024) from $0.436$ to $0.705$ on box inpainting.
- We empirically show that DiffStateGrad improves the worst-case performance of the diffusion model, e.g., significantly reducing the failure rate (PSNR $< 20$) from $26\%$ to $4\%$ on the phase retrieval task, increasing their reliability (Figure 3). DiffStateGrad consistently shows lower standard deviation across the test datasets than state-of-the-art methods.
- We demonstrate that DiffStateGrad significantly improves the performance of state-of-the-art (SOTA) methods, especially in challenging tasks such as phase retrieval and high dynamic range reconstruction. For example, DiffStateGrad improves the PSNR of ReSample (Song et al., 2023a) from $27.61(8.07)$ to $31.19(4.33)$ for phase retrieval, reporting mean (std). Our experiments cover a wide range of linear inverse problems of box inpainting, random inpainting, Gaussian deblur, motion deblur, and super-resolution (Tables 3 and 4) and nonlinear inverse problems of phase retrieval, nonlinear deblur, and high dynamic range (HDR) (Table 3) for image restoration tasks.

## 2 BACKGROUND & RELATED WORKS

**Learning-based priors.** These methods leverage data structures captured by a pre-trained denoiser (Romano et al., 2017) as plug-and-play priors (Venkatakrishnan et al., 2013), or deep generative models such as variational autoencoders (VAEs) (Kingma, 2013) and generative adversarial networks (GANs) (Goodfellow et al., 2014) to solve inverse problems (Bora et al., 2017; Ulyanov et al., 2018). The state-of-the-art is based on generative diffusion models, which have shown promising performance in generating high-quality samples in computer vision (Song

| Projection Subspace | LPIPS↓ | SSIM↑ | PSNR↑ |
|---|---|---|---|
| No Projection | 0.246 | 0.809 | 29.05 |
| Random matrix | 0.299 | 0.753 | 27.30 |
| Measurement gradient | 0.242 | 0.808 | 29.21 |
| DiffStateGrad (ours) | **0.165** | **0.898** | **31.68** |

Table 1: **Advantage of diffusion state-guided projection.** Results are from random inpainting on FFHQ $256 \times 256$.

et al., 2023b), solving PDEs (Shu et al., 2023), and high-energy physics (Shmakov et al., 2024).

**Diffusion models.** Diffusion models conceptualize the generation of data as the reverse of a noising process, where a data sample $x_t$ at time $t$ within the interval $[0, T]$ follows a specified stochastic differential equation (SDE). This SDE (Song et al., 2021) for the data noising process is described by

$$d\boldsymbol{x} = -(\beta_t/2)\boldsymbol{x}\, dt + \sqrt{\beta_t}\, d\boldsymbol{w}, \tag{2}$$

where $\beta_t \in (0, 1)$ is a positive, monotonically increasing function of time $t$, and $\boldsymbol{w}$ represents a standard Wiener process. The process begins with an initial data distribution $\boldsymbol{x}_0 \sim p_{\text{data}}$ and transitions to an approximately Gaussian distribution $\boldsymbol{x}_T \sim \mathcal{N}(\boldsymbol{0}, \boldsymbol{I})$ by time $T$. The objective of regenerating the original data distribution from this Gaussian distribution involves reversing the

noising process through a reverse SDE of the form

$$dx = [-(\beta_t/2)x - \beta_t \nabla_{x_t} \log p_t(x_t)]dt + \sqrt{\beta_t}d\bar{w}, \tag{3}$$

where $dt$ indicates time moving backward and $\bar{w}$ is the reversed Wiener process. To approximate $\nabla_{x_t} \log p_t(x_t)$, a neural network $s_\theta$ trained via denoising score matching (Vincent, 2011) is used.

**Solving inverse problems with diffusion models.** Diffusion-based approaches to inverse problems seek to reconstruct the data $x_0$ from the measurement $y = \mathcal{A}(x_0) + n$ via the following reverse SDE

$$dx = [-(\beta_t/2)x - \beta_t(\nabla_{x_t} \log p_t(x_t) + \nabla_{x_t} \log p_t(y|x_t))]dt + \sqrt{\beta_t}d\bar{w}. \tag{4}$$

Conceptually, the learned score function $\nabla_{x_t} \log p_t(x_t)$ guides the reverse diffusion process from noise to the data distribution, and the likelihood-related term $\nabla_{x_t} \log p_t(y \mid x_t)$ ensures measurement consistency. When the model is trained unconditionally, the main challenge is the intractable denoiser posterior due to the lack of an explicit analytical expression for $\nabla_{x_t} \log p_t(y \mid x_t)$; the exact relationship between $y$ and intermediate states $x_t$ is not well-defined, except at the initial state $x_0$.

**Solving inverse problems with latent diffusion models.** For complex scenarios where direct application of pixel-based models is computationally expensive or ineffective, latent diffusion models (LDMs) offer a promising alternative (Rombach et al., 2022). Given data $x \in \mathbb{R}^n$, the LDM framework utilizes an encoder $\mathcal{E} : \mathbb{R}^n \to \mathbb{R}^k$ and a decoder $\mathcal{D} : \mathbb{R}^k \to \mathbb{R}^n$, with $k \ll n$, to work in a compressed latent space. $x_T$ is encoded into a latent representation $z_T = \mathcal{E}(x_T)$ and serves as the starting point for the reverse diffusion process. Then, $z_0$ is decoded to $x_0 = \mathcal{D}(z_0)$, the final image reconstruction. Using a *latent* diffusion model introduces an additional complexity to solving inverse problems. The challenge arises from the nonlinear nature and non-uniqueness mapping of the encoder/decoder Rout et al. (2023); PSLD proposed to improve performance by enforcing fixed-point properties on representations.

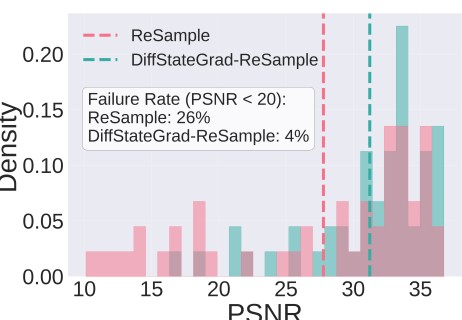

Figure 3: **DiffStateGrad improves the worst-case performance**. The PSNR histogram for phase retrieval shows that DiffStateGrad significantly lowers the failure rate.

**Diffusion-based inverse problems addressing challenges of intractable denoiser posterior.** To address the intractability of the gradient for the reverse diffusion, Diffusion Posterior Sampling (DPS) (Chung et al., 2023), approximates the probability $p(y \mid x_t) \approx p(y|\hat{x}_0 := \mathbb{E}[x_0 \mid x_t])$ using the conditional expectation of the data. Extending to the latent case, *Latent-DPS* uses $p(y|z_t) \approx p(y|\hat{x}_0 := \mathcal{D}(\mathbb{E}[z_0|z_t]))$ (Song et al., 2023a). Two intuitive drawbacks of this approach are that a) the image estimate $\hat{x}_0$ is reconstructed using an expectation, which results in inaccurate estimations for multi-modal complex distributions, and b) the measurement gradient directly updates the noisy state $x_t$, which may push away the state from the desired noise level at $t$.

Prior works aim to address the first challenge by going beyond first-order statistics (Rout et al., 2024) or incorporating posterior covariance into the maximum likelihood estimation step (Peng et al., 2024). Other lines of work address the second issue by decoupling the measurement guidance from the sampling process; they update the data estimate $\hat{x}_0$ at time $t$ using the measurement gradient guidance before resampling it to the noisy manifold at time $t - 1$ (Song et al., 2023a; Zhang et al., 2024). The above-discussed approaches are still highly sensitive to the measurement gradient step size (Peng et al., 2024). Indeed, balancing the measurement gradient with the unconditional score function remains a significant challenge to solving inverse problems using measurement-guided generation. Wu et al. (2024) avoids the discussed approximations and samples from the posterior directly to resolve the need to find a balance between measurement guidance and the prior process.

**Projections in diffusion models.** Manifold and subspace projections are used in various contexts in diffusion models. MPGD (He et al., 2024) uses a manifold-preserving approach to improve the efficiency of diffusion generation and solving inverse problems. While this method follows a similar sentiment as our proposed framework, it is only applicable when the measurement gradient is applied to $\hat{x}_{0|t}$. Moreover, it requires the existence of an autoencoder for achieving manifold projection, and its performance is heavily dependent on the expressive power of the autoencoder. Unlike MPGD, DiffStateGrad is applicable to methods that apply the guidance to $\hat{x}_{0|t}/\hat{z}_{0|t}$ (i.e., ReSample and

DAPS) and methods that apply the guidance to $\hat{x}_t/\hat{z}_t$ (i.e., PSLD and DPS). Chung et al. (2022a) proposes a manifold constraint to project the measurement gradient into the data manifold $x_0$ while the guidance updates $x_t$; our proposal is more effective since we project the measurement gradient on the noisy diffusion state related to $x_t$ or $z_t$, preserving the $t$ state on $\mathcal{M}_t$ rather than $\mathcal{M}_0$.

**Gradient guidance incorporation.** Prior works differ from one another in two key aspects: (a) how the loss for the gradient is computed and (b) how the gradient is used to update the diffusion state. Table 2 categorizes prior works based on these characteristics. While a few approaches, such as diffusion-based MRI (Chung & Ye, 2022), compute the gradient using $x_t$, most recent literature has shifted to-

| Method | Gradient computation | Gradient incorporation | | Projection | |
|---|---|---|---|---|---|
| | $x_{0\|t}/z_{0\|t}$ | $x_t/z_t$ | $x_{0\|t}/z_{0\|t}$ | $x_t/z_t$ | $x_{0\|t}/z_{0\|t}$ |
| DPS | ✓ | ✓ | ✗ | ✗ | ✗ |
| PSLD | ✓ | ✓ | ✗ | ✗ | ✗ |
| ReSample | ✓ | ✗ | ✓ | ✗ | ✗ |
| DAPS | ✓ | ✗ | ✓ | ✗ | ✗ |
| MCG | ✓ | ✓ | ✗ | ✗ | ✓ |
| MPGD | ✓ | ✗ | ✓ | ✗ | ✓ |
| DiffStateGrad (ours) | ✓ | ✓ | ✓ | ✓ | ✓ |

Table 2: **Gradient guidance computation, incorporation, and projection for diffusion-based inverse problems.** ward using $x_{0\|t}$ for gradient computation. Regarding gradient incorporation, the literature is further subdivided. For instance, methods like DPS and PSLD use the measurement gradient to update the state at time $t$, whereas ReSample, DAPS, and MPGD apply the guidance to the conditional state at $0\|t$ before resampling. Additionally, while the projections in MCG (Chung et al., 2022a) and MPGD (He et al., 2024) are restricted to $x_{0\|t}$, DiffStateGrad applies to both types of methods.

**Conditional diffusion models for inverse problems.** We focus on unconditional diffusion models as learned priors to solve general inverse problems. This approach leverages *already trained* diffusion models, which is useful for domains with abundant data. Another approach is to train conditional diffusion models where $\nabla_{x_t} \log p_t(y \mid x_t)$ is directly captured by the score function, or where the diffusion directly transforms the measurement into the underlying data (e.g., image-to-image diffusion) (Saharia et al., 2022; Liu et al., 2023; Chung et al., 2024). This latter approach is problem-specific; hence, it is not generalizable across inverse tasks. Finally, we note that while this work focuses on gradient-based guidance, prior work such as RePaint (Lugmayr et al., 2022) introduces a gradient-free masking strategy to solve inverse problems. Although RePaint is appealing, it is limited to inpainting tasks and scenarios where measurement noise is negligible.

## 3 DIFFUSION STATE-GUIDED PROJECTED GRADIENT (DIFFSTATEGRAD)

We propose a *Diffusion State-Guided Projected Gradient* (DiffStateGrad) to solve inverse problems. DiffStateGrad can be incorporated into a wide range of diffusion models to improve guidance-based diffusion models. Without loss of generality, we explain DiffStateGrad in the context of Latent-DPS (Chung et al., 2023) hich applies the measurement guidance to $z_t$. We note that DiffStateGrad applies to a wide range of pixel and latent diffusion-based inverse solvers (see Section 4).

Given $z_{t+1}$, we sample $z_t$ from the unconditional reverse process, and then compute the estimate $\hat{z}_0(z_t) := \mathcal{D}(\mathbb{E}[z_0 \mid z_t])$. Then, the data-consistency guidance term can be incorporated as follows.

$$z_t \leftarrow z_t - \eta_t \mathcal{P}_{\mathcal{S}_t}(g_t), \tag{5}$$

where $g_t = \nabla_{z_{t+1}} \log p(y \mid \hat{z}_0(z_t))$ is the measurement gradient (MG), $\eta_t$ is the step size, and $\mathcal{P}_{\mathcal{S}_t}$ is a projection step onto the low-rank subspace $\mathcal{S}_t$. The main contribution of this paper is a) to highlight that the measurement gradient should be projected onto a subspace imposed by the state being updated by the gradient (see gradient incorporation column in Table 2) and b) to define this subspace so it results in better posterior sampling; in other words, to define a subspace such that when the measurement gradient is projected onto, the diffusion process is not disturbed and pushed away from the data manifold. In Table 1, we show for PSLD that indeed the subspace $\mathcal{S}_t$, defined by the intermediate diffusion state, results in an improved posterior sampling, unlike a subspace that is constructed based on a random matrix or the low-rank structure of the measurement gradient. Hence, we choose the diffusion state $z_t$ to define $\mathcal{S}_t$. Finally, for methods where the measurement gradient guidance is being applied to $z_{0\|t}$, we define the low-rank subspace based on $z_{0\|t}$ (see Table 2).

We focus on images as our data modality and implement the projection $\mathcal{P}_{\mathcal{S}_t}$ by computing the SVD of $z_t$ in its image matrix form, denoted by $Z_t$ (i.e., $U, S, V \leftarrow \text{SVD}(Z_t)$). Then, we compute an adaptive rank $r \leftarrow \arg\min_k \{\sum_{j=1}^k s_j^2 / \sum_j s_j^2 \geq \tau\}$ leveraging a fixed variance retention threshold $\tau$.

---

**Algorithm 1** Diffusion State-Guided Projected Gradient (DiffStateGrad) for Latent Diffusion-based Inverse Problems (Image Restoration Tasks)

---

**Require:** Normal input + variance retention threshold $\tau$
 1: Let $T$ = number of total iterations of sampling algorithm and assume we calculate latent image representation $\boldsymbol{Z}_t$ for each iteration. Note that $\boldsymbol{Z}_t$ is a matrix.
 2: **for** $t = T - 1$ **to** $0$ **do**
 3:     Compute measurement gradient $\boldsymbol{G}_t$ according to sampling algorithm
 4:     $\boldsymbol{U}, \boldsymbol{S}, \boldsymbol{V} \leftarrow \text{SVD}(\boldsymbol{Z}_t)$                      ▷ Perform SVD on current diffusion state
 5:     $\lambda_j \leftarrow s_j^2$ (where $s_j$ are the singular values of $\boldsymbol{S}$)             ▷ Calculate eigenvalues
 6:     $c_k \leftarrow \frac{\sum_{j=1}^{k} \lambda_j}{\sum_j \lambda_j}$                              ▷ Cumulative sum of eigenvalues
 7:     $r \leftarrow \underset{k}{\arg\min}\{c_k \geq \tau\}$                    ▷ Determine rank $r$ based on threshold $\tau$
 8:     $\boldsymbol{A}_t \leftarrow \boldsymbol{U}_r$                             ▷ Get first $r$ left singular vectors
 9:     $\boldsymbol{B}_t \leftarrow \boldsymbol{V}_r$                            ▷ Get first $r$ right singular vectors
10:     $\boldsymbol{R}_t \leftarrow \boldsymbol{A}_t^T \boldsymbol{G}_t \boldsymbol{B}_t^T$                          ▷ Project gradient
11:     $\boldsymbol{G}_t' \leftarrow \boldsymbol{A}_t \boldsymbol{R}_t \boldsymbol{B}_t$                  ▷ Reconstruct approximated gradient
12:     Use updated gradient $\boldsymbol{G}_t'$ in sampling algorithm
13: **end for**
14: **return** $\mathcal{D}(\hat{\boldsymbol{z}}_0)$

---

The gradient $\boldsymbol{g}_t$, which takes a matrix form for images, is projected onto a subspace defined by the highest $r$ singular values of $\boldsymbol{Z}_t$ as follows:

$$\boldsymbol{G}_t \leftarrow \boldsymbol{U}_r \boldsymbol{U}_r^{\mathrm{T}} \boldsymbol{G}_t \boldsymbol{V}_r^{\mathrm{T}} \boldsymbol{V}_r, \tag{6}$$

where $\boldsymbol{G}_t$ is the measurement gradient in image matrix form, and $\boldsymbol{U}_r$ and $\boldsymbol{V}_r$ contain the first $r$ left and right singular vectors, respectively (Section 3). While we use the *full* SVD projection (i.e., combining both left and right projection), in practice, one may choose to do either left or right projection. Next, we provide mathematical intuitions (Proposition 1) on the effectiveness of subspace projections in preserving $\boldsymbol{z}_t$, particularly for high-dimensional data with low-rank structure, after the MG update on the manifold $\mathcal{M}_t$. Finally, we note that while DiffStateGrad can significantly improve the runtime and computational efficiency of diffusion frameworks that use Adam optimizers for data consistency (Song et al., 2023a; Zhao et al., 2024), the current implementation and this paper does not explore this aspect and, instead, focuses on the property of the proposed subspace.

**Proposition 1.** *Let $\mathcal{M}$ be a smooth $m$-dimensional submanifold of a $d$-dimensional Euclidean space $\mathbb{R}^d$, where $m < d$. Assume that for each state $\boldsymbol{z}_t \in \mathcal{M}$, the tangent space $T_{\boldsymbol{z}_t}\mathcal{M}$ is well-defined, and the projection operator $\mathcal{P}_{\mathcal{S}_{\boldsymbol{z}_t}}$ onto an approximate subspace $\mathcal{S}_{\boldsymbol{z}_t}$ closely approximates the projection onto $T_{\boldsymbol{z}_t}\mathcal{M}$. For the state $\boldsymbol{z}_t \in \mathcal{M}$ and measurement gradient $\boldsymbol{g}_t \in \mathbb{R}^d$, consider two update rules:*

$$\begin{aligned}
\boldsymbol{z}_{t-1} &= \boldsymbol{z}_t - \eta \boldsymbol{g}_t \quad \text{(standard update)}, \\
\boldsymbol{z}_{t-1}' &= \boldsymbol{z}_t - \eta \mathcal{P}_{\mathcal{S}_{\boldsymbol{z}_t}}(\boldsymbol{g}_t) \quad \text{(projected update)},
\end{aligned} \tag{7}$$

*where $\eta > 0$ is a small step size. Then, for sufficiently small $\eta$, the projected update $\boldsymbol{z}_{t-1}'$ stays closer to the manifold $\mathcal{M}$ than the standard update $\boldsymbol{z}_{t-1}$. That is,*

$$dist(\boldsymbol{z}_{t-1}', \mathcal{M}) < dist(\boldsymbol{z}_{t-1}, \mathcal{M}). \tag{8}$$

The remainder of this section provides intuition on how DiffStateGrad improves solving inverse problems in the presence of a suitable learned prior. Let the initial latent state $\boldsymbol{z}_t$ be on the manifold $\mathcal{M}_t$ (e.g., being artifact-free). The term "artifact-free" refers to the generation process of an unconditional diffusion model that is trained on clean data samples and provides an artifact-free trajectory from $\mathcal{M}_T$ to $\mathcal{M}_0$. We observe that pushing away from the manifold process (e.g., introducing artifacts) can only be introduced via the guidance by the data-consistency gradient step, as this is the sole mechanism by which information from the measurement process enters the latent space. Consider the manifold $\mathcal{M}$ of artifact-free latent representations. Each $\boldsymbol{z}_t$ lies on this manifold, and the tangent space $T_{\boldsymbol{z}_t}\mathcal{M}$ represents the directions of "allowable" updates that maintain the artifact-free property staying on the current manifold. Finally, we note that this motivates to project the gradient onto the tangent space of the data manifold where the guidance is applied. Alternatively, when the measurement gradient guidance is applied to $\boldsymbol{z}_{0|t}$, we define the projection step based on $\mathcal{S}_0$ (see also (He et al., 2024)).

Our DiffStateGrad method, through the projection operator $\mathcal{P}_{\mathcal{S}_t}$, approximates this tangent space. The effectiveness of DiffStateGrad depends on how well $\mathcal{P}_{\mathcal{S}_t}$ approximates the projection onto $T_{\boldsymbol{z}_t}\mathcal{M}$. Hence, we discuss a rationale on how the approximated projection is sufficient for performance; we, accordingly, support this by experimental results in Section 4. First, the SVD captures the principal directions of variation in $\boldsymbol{z}_t$, which are likely to align with the local structure of the manifold when the data is high-dimensional. Second, by adaptively choosing the rank based on a variance retention threshold, we ensure that the projection preserves the most significant state-related structural information while filtering out potential noise or artifact-inducing components from the measurement gradient. Finally, the low-rank nature of our approximation aligns with the assumption that the manifold of representations has a lower intrinsic dimensionality than the ambient space.

Hence, by projecting the measurement gradient onto this subspace defined by the current latent state $\boldsymbol{z}_t$, we effectively filter the directions orthogonal to the local manifold structure, and hence, remove artifacts-inducing components. This projection ensures that updates to $\boldsymbol{z}_t$ remain closer to the manifold $\mathcal{M}$ than unprojected updates would, as stated in Proposition 1. Consequently, DiffStateGrad relies on the reliability of the learned prior and helps to provide high-probability posterior samples. This creates an inductive process: if $\boldsymbol{z}_t$ is artifact-free, and we only allow updates that align with its structure (i.e., updates that stay close to the manifold $\mathcal{M}$), subsequent latent representation $\boldsymbol{z}_t$ will likely be samples from the high-probability regions of the posterior.

Figure 2 demonstrates the effectiveness of DiffStateGrad in removing artifacts when the MG step size is large; artifacts are introduced onto the measurement gradient and stay within the latent representation in PSLD (Rout et al., 2023). On the other hand, the reverse process via DiffStateGrad-PSLD (our method applied to PSLD) stays artifact-free, consistent with the mathematical analysis in Proposition 1 and the practical efficacy of the proposed SVD-based subspace projection. Finally, we note that the most significant improvements appear in challenging tasks such as phase retrieval, HDR, and inpainting. We attribute the effectiveness of DiffStateGrad, particularly in challenging tasks, to a reduced rate of failure cases (Figure 3). By constraining solutions closer to the data manifold, DiffStateGrad minimizes extreme failures, enhances consistency in reconstruction quality.

**Efficiency.** DiffStateGrad introduces minimal computational overhead. We perform SVD *at most* once per iteration, and for latent diffusion solvers, this occurs in the latent space on $64 \times 64$ matrices. By selecting a low rank based on a variance threshold, subsequent projection and reconstruction operations are performed on reduced matrices, further decreasing computational complexity. Figure 4 illustrates the runtime of PSLD (Rout et al., 2023), ReSample (Song et al., 2023a), and DAPS (Zhang et al., 2024) with and without DiffStateGrad. The figure shows

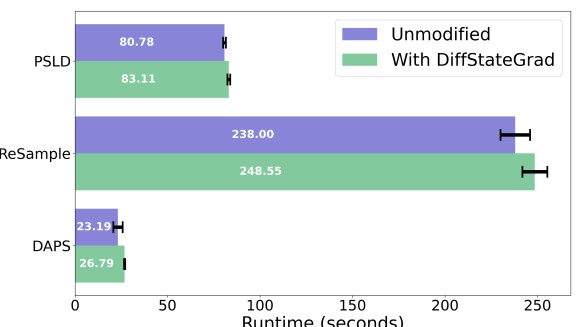

Figure 4: **Runtime complexity of DiffStateGrad.** The increase of runtime with DiffStateGrad is minimal.

that the additional computational cost of incorporating DiffStateGrad is marginal, typically adding only a few seconds to the total runtime (see C.2 for further details). We note that our method is not intended to *improve* efficiency, but rather to enhance performance and robustness.

## 4 RESULTS

This section provides extensive experimental results on the effectiveness of DiffStateGrad for image-based inverse problems. We show that DiffStateGrad significantly improves (1) the robustness of diffusion-based methods to the choice of measurement gradient step size and measurement noise, and (2) the overall posterior sampling performance of diffusion.

**Experimental setup.** We evaluate the performance of DiffStateGrad applied to four SOTA diffusion methods of PSLD (Rout et al., 2023), ReSample (Song et al., 2023a), DPS (Chung et al., 2023), and DAPS (Zhang et al., 2024). These methods span both latent solvers (PSLD and ReSample) and pixel-based solvers (DPS and DAPS). We also directly compare against other methods including DDNM (Wang et al., 2023), DDRM (Kawar et al., 2022), MCG (Chung et al., 2022a), and

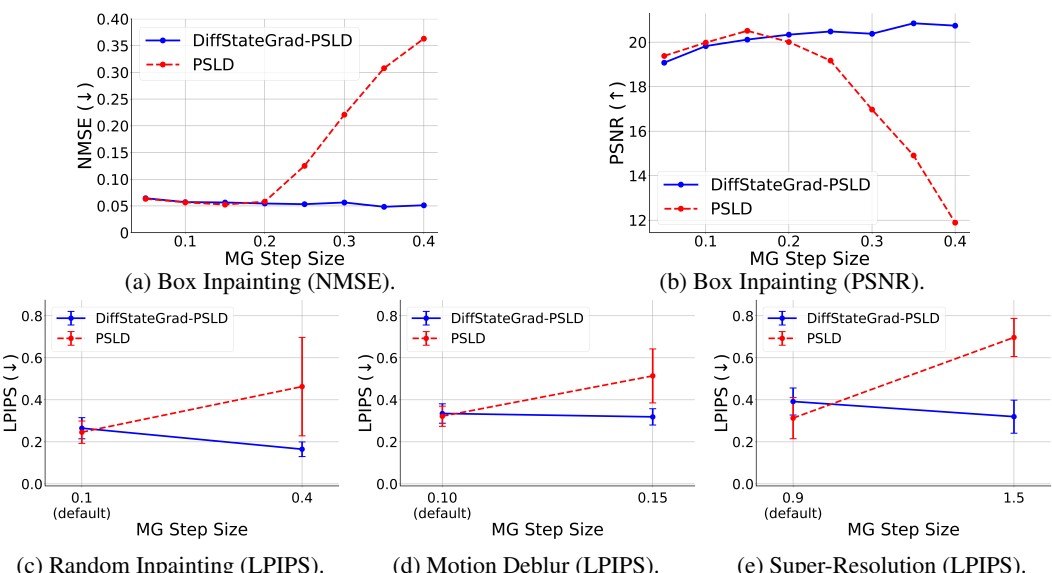

(a) Box Inpainting (NMSE).  (b) Box Inpainting (PSNR).

(c) Random Inpainting (LPIPS).  (d) Motion Deblur (LPIPS).  (e) Super-Resolution (LPIPS).

Figure 5: **Robustness of DiffStateGrad to MG step size**. (a-b) Performance on box inpainting across various MG step sizes. (c-e) Performance on different tasks with default and large step sizes. We evaluate the performance of PSLD and DiffStateGrad-PSLD using FFHQ $256 \times 256$.

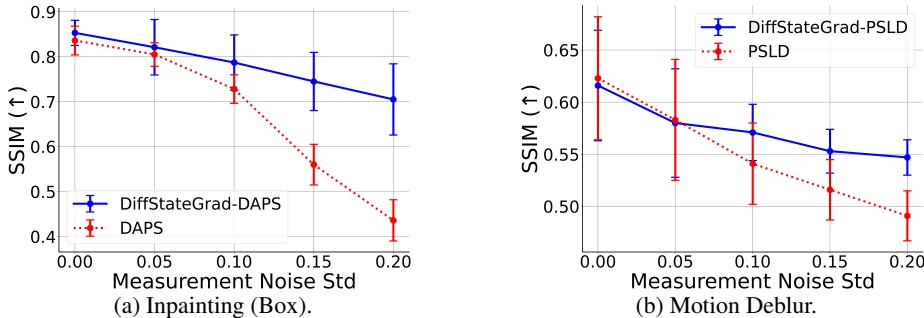

(a) Inpainting (Box).  (b) Motion Deblur.

Figure 6: **Robustness of DiffStateGrad to measurement noise.** We evaluate the performance of DiffStateGrad-PSLD and DiffStateGrad-DAPS with their respective counterparts on different tasks across a range of measurement noise levels (std of 0 to 0.2) using FFHQ $256 \times 256$.

MPGD-AE (He et al., 2024). We evaluate performance based on key quantitative metrics, including LPIPS (Learned Perceptual Image Patch Similarity), PSNR (Peak Signal-to-Noise Ratio), and SSIM (Structural Similarity Index) (Wang et al., 2004). We demonstrate the effectiveness of DiffStateGrad on two datasets: a) the FFHQ $256 \times 256$ validation dataset (Karras et al., 2021), and b) the ImageNet $256 \times 256$ validation dataset (Deng et al., 2009). For pixel-based experiments, we use (i) the pre-trained diffusion model from (Chung et al., 2023) for the FFHQ dataset, and (ii) the pre-trained model from (Dhariwal & Nichol, 2021) for the ImageNet dataset. For latent diffusion experiments, we use (i) the unconditional LDM-VQ-4 model trained on FFHQ (Rombach et al., 2022) for the FFHQ dataset, and (ii) the Stable Diffusion v1.5 (Rombach et al., 2022) model for the ImageNet dataset.

We consider both linear and nonlinear inverse problems for natural images. For evaluation, we sample a fixed set of 100 images from the FFHQ and ImageNet validation sets. Images are normalized to the range $[0, 1]$. We use the default settings for all experiments (see Appendix C for more details). For linear inverse problems, we consider (1) box inpainting, (2) random inpainting, (3) Gaussian deblur, (4) motion deblur, and (5) super-resolution. In the box inpainting task, a random $128 \times 128$ box is used, while the random inpainting task employs a $70\%$ random mask. Gaussian and motion deblurring tasks utilize kernels of size $61 \times 61$, with standard deviations of $3.0$ and $0.5$, respectively. For super-resolution, images are downscaled by a factor of $4$ using a bicubic resizer. For nonlinear inverse problems, we consider (1) phase retrieval, (2) nonlinear deblur, and (3) high dynamic range (HDR). For phase retrieval, we use an oversampling rate of $2.0$, and due to the instability and non-uniqueness of reconstruction, we adopt the strategy from DPS (Chung et al., 2023) and DAPS (Zhang

Table 3: **Performance comparison for linear and nonlinear tasks on FFHQ 256 × 256.**

| Method | Inpaint (Box) | | Inpaint (Random) | | Gaussian deblur | | Motion deblur | | SR (x4) | |
|---|---|---|---|---|---|---|---|---|---|---|
| | LPIPS↓ | PSNR↑ | LPIPS↓ | PSNR↑ | LPIPS↓ | PSNR↑ | LPIPS↓ | PSNR↑ | LPIPS↓ | PSNR↑ |
| *Pixel-based* | | | | | | | | | | |
| DAPS | 0.136 | 24.57 | 0.130 | 30.79 | 0.216 | 27.92 | 0.154 | 30.13 | 0.197 | 28.64 |
| DiffStateGrad-DAPS (ours) | **0.113** | **24.78** | **0.099** | **32.04** | 0.180 | **29.02** | 0.119 | 31.74 | 0.181 | **29.35** |
| DPS | 0.127 | 23.91 | 0.130 | 28.67 | 0.145 | 25.48 | 0.132 | 26.75 | 0.191 | 24.38 |
| DiffStateGrad-DPS (ours) | 0.114 | 24.10 | 0.107 | 30.15 | **0.128** | 26.29 | **0.118** | 27.61 | 0.186 | 24.65 |
| DDNM | 0.235 | 24.47 | 0.121 | 29.91 | 0.216 | 28.20 | - | - | 0.197 | 28.03 |
| DDRM | 0.159 | 22.37 | 0.218 | 25.75 | 0.236 | 23.36 | - | - | 0.210 | 27.65 |
| MCG | 0.309 | 19.97 | 0.286 | 21.57 | 0.340 | 6.72 | 0.702 | 6.72 | 0.520 | 20.05 |
| MPGD-AE | 0.138 | 21.59 | 0.172 | 25.22 | 0.150 | 24.42 | 0.120 | 25.72 | **0.168** | 24.01 |
| *Latent* | | | | | | | | | | |
| PSLD | 0.158 | 24.22 | 0.246 | 29.05 | 0.357 | 22.87 | 0.322 | 24.25 | 0.313 | 24.51 |
| DiffStateGrad-PSLD (ours) | **0.092** | **24.32** | 0.165 | 31.68 | 0.355 | 22.95 | 0.319 | 24.31 | 0.320 | 24.56 |
| ReSample | 0.198 | 19.91 | 0.115 | 31.27 | 0.253 | 27.78 | 0.160 | 30.55 | 0.204 | 28.02 |
| DiffStateGrad-ReSample (ours) | 0.156 | 23.59 | **0.106** | **31.91** | **0.245** | **28.04** | **0.153** | **30.82** | **0.200** | **28.27** |

(a) Linear inverse problems.

| Method | Phase retrieval | | Nonlinear deblur | | High dynamic range | |
|---|---|---|---|---|---|---|
| | LPIPS↓ | PSNR↑ | LPIPS↓ | PSNR↑ | LPIPS↓ | PSNR↑ |
| *Pixel-based* | | | | | | |
| DAPS | 0.139 (0.026) | 30.52 (2.61) | 0.184 (0.032) | 27.80 (1.97) | 0.170 (0.075) | 26.91 (3.94) |
| DiffStateGrad-DAPS (ours) | **0.105** (0.023) | **32.25** (1.34) | **0.145** (0.027) | **29.51** (2.16) | **0.143** (0.070) | **27.76** (3.18) |
| *Latent* | | | | | | |
| ReSample | 0.237 (0.189) | 27.61 (8.07) | 0.188 (0.037) | 29.54 (1.89) | 0.190 (0.067) | 24.88 (3.46) |
| DiffStateGrad-ReSample (ours) | **0.154** (0.104) | **31.19** (4.33) | **0.185** (0.035) | **29.91** (1.60) | **0.164** (0.041) | **25.50** (3.07) |

(b) Nonlinear inverse problems.

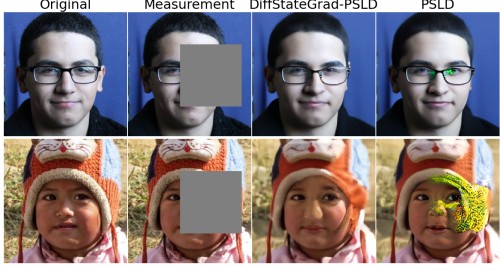

(a) Box inpainting.

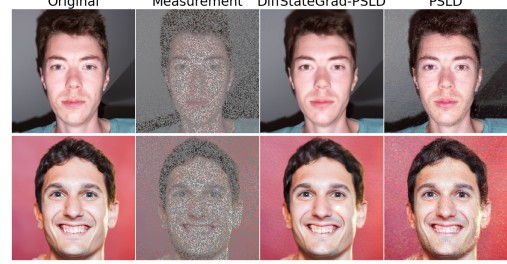

(b) Random inpainting.

Figure 7: **Qualitative comparison of DiffStateGrad-PSLD and PSLD for their best-performing MG step size**. DiffStateGrad-PSLD can remove artifacts and reduce failure cases, producing more reliable reconstructions. Images are chosen at random for visualization.

et al., 2024), generating four separate reconstructions and reporting the best result. Like DAPS (Zhang et al., 2024), we normalize the data to lie in the range [0, 1] before applying the discrete Fourier transform. For nonlinear deblur, we use the default setting from (Tran et al., 2021). For HDR, we use a scale factor of 2. We note that PSLD is not designed to handle nonlinear inverse problems. We also conduct an additional experiment for Magnetic Resonance Imaging (MRI) (see Appendix E).

**Robustness.** Figure 5 exhibits the sensitivity of PSLD to the choice of MG step size; the performance of PSLD significantly deteriorates when a relatively large MG step size is used, leading to poor results across all tasks. In contrast, DiffStateGrad-PSLD shows superior robustness, maintaining high performance over a wide range of MG step sizes. Figure 6 demonstrates the robustness of DiffStateGrad methods compared to their non-DiffStateGrad counterparts when faced with increasing measurement noise. For both inpainting and Gaussian deblur tasks, the performance of DAPS and PSLD deteriorates significantly as noise levels rise. In contrast, DiffStateGrad-DAPS and DiffStateGrad-PSLD exhibit superior resilience across the range of noise levels tested.

Table 4: **Performance comparison for linear tasks on ImageNet 256 × 256.**

| Method | Inpaint (Box) | | | Inpaint (Random) | | |
|---|---|---|---|---|---|---|
| | LPIPS↓ | SSIM↑ | PSNR↑ | LPIPS↓ | SSIM↑ | PSNR↑ |
| *Pixel-based* | | | | | | |
| DAPS | 0.217 (0.043) | 0.762 (0.041) | 20.90 (3.69) | 0.158 (0.039) | 0.794 (0.067) | 28.34 (3.65) |
| DiffStateGrad-DAPS (ours) | **0.191** (0.044) | **0.801** (0.056) | **21.07** (3.77) | **0.107** (0.037) | **0.856** (0.067) | **29.78** (4.17) |
| DPS | 0.257 (0.086) | 0.718 (0.097) | 19.85 (3.54) | 0.256 (0.133) | 0.728 (0.143) | 26.25 (4.15) |
| DiffStateGrad-DPS (ours) | 0.243 (0.093) | 0.731 (0.100) | 19.87 (3.61) | 0.233 (0.138) | 0.754 (0.150) | 27.28 (4.88) |
| MPGD-AE | 0.295 (0.057) | 0.621 (0.053) | 16.12 (2.26) | 0.554 (0.148) | 0.388 (0.112) | 17.91 (3.25) |
| *Latent* | | | | | | |
| PSLD | 0.182 (0.033) | 0.780 (0.044) | 16.28 (3.49) | 0.217 (0.073) | 0.846 (0.070) | 26.56 (2.98) |
| DiffStateGrad-PSLD (ours) | **0.176** (0.030) | **0.803** (0.045) | **18.90** (3.82) | **0.169** (0.050) | **0.878** (0.051) | **28.48** (4.04) |

(a) Inpainting inverse problems.

| Method | Gaussian deblur | | Motion deblur | | SR (x4) | |
|---|---|---|---|---|---|---|
| | LPIPS↓ | PSNR↑ | LPIPS↓ | PSNR↑ | LPIPS↓ | PSNR↑ |
| *Pixel-based* | | | | | | |
| DAPS | 0.266 (0.087) | 25.27 (3.56) | 0.166 (0.058) | 28.85 (3.64) | 0.259 (0.073) | 25.67 (3.40) |
| DiffStateGrad-DAPS (ours) | **0.243** (0.075) | **25.87** (3.56) | **0.143** (0.050) | **29.71** (3.54) | **0.229** (0.057) | **26.40** (3.44) |
| MCG | 0.550 (-) | 16.32 (-) | 0.758 (-) | 5.89 (-) | 0.637 (-) | 13.39 (-) |
| *Latent* | | | | | | |
| PSLD | 0.466 (0.085) | 20.70 (3.01) | 0.435 (0.102) | 21.26 (3.44) | 0.416 (0.063) | 22.29 (3.08) |
| DiffStateGrad-PSLD (ours) | **0.446** (0.076) | **22.34** (3.19) | **0.399** (0.060) | **23.80** (3.27) | **0.370** (0.081) | **23.53** (3.52) |

(b) Deblurring and super-resolution inverse problems.

**Performance.** We provide quantitative results in Tables 3 to 4, and qualitative results in Figure 7. These results demonstrate the substantial improvement in the performance of the methods with DiffStateGrad against their respective SOTA counterparts across a wide variety of linear and nonlinear tasks on both FFHQ and ImageNet. For example, DiffStateGrad significantly improves performance and increases reconstruction consistency in both pixel-based and latent solvers for phase retrieval (Table 3). Our results also highlight the superiority of our choice of subspace compared to prior works. For example, DiffStateGrad-DPS is superior to MCG, which is the special case of DPS for manifold constrained diffusion. Furthermore, the outperformance of DiffStateGrad-DAPS against DAPS and MPGD-AE emphasizes the effectiveness of the proposed SVD-based subspace projection. We additionally show that DiffStateGrad does not impact the diversity of the posterior sampling (Figure 16). Finally, we note that DiffStateGrad is robust to the choice of subspace rank (Figures 8 and 9). We refer the reader to Appendix B for the extensive qualitative performance of DiffStateGrad.

## 5 CONCLUSION

We introduce a *Diffusion State-Guided Projected Gradient* (DiffStateGrad) to enhance the performance and robustness of diffusion models in solving inverse problems. DiffStateGrad addresses the introduction of artifacts and deviations from the data manifold by constraining gradient updates to a subspace approximating the manifold. DiffStateGrad is versatile, applicable across various diffusion models and sampling algorithms, and includes an adaptive rank that dynamically adjusts to the gradient's complexity. Overall, DiffStateGrad reduces the need for excessive tuning of hyperparameters and significantly boosts performance for more challenging inverse problems. We note that DiffStateGrad assumes that the learned prior is a relatively good prior for the task at hand. Since DiffStateGrad encourages the process to stay close to the manifold structure captured by the generative prior, it may introduce the prior's biases into image restoration tasks. Hence, DiffStateGrad may not be recommended for certain inverse problems such as black hole imaging (Feng et al., 2024). We finally note that DiffStateGrad can be combined with prior works that adopt initialization strategies for the diffusion process to further accelerate and improve performance (Fabian et al., 2024; Chung et al., 2022b). We leave this for future work.

## 6 REPRODUCIBILITY STATEMENT

To ensure the reproducibility of our results, we thoroughly detail the hyperparameters used in our experiments in C.1. We also provide specific implementation and configuration details of all the baselines used in C.3, C.4, C.5, and C.6. Moreover, we use easily accessible pre-trained diffusion models throughout our experiments. PSLD (`https://github.com/LituRout/PSLD`), ReSample (`https://github.com/soominkwon/resample`), DPS (`https://github.com/DPS2022/diffusion-posterior-sampling`), and DAPS (`https://github.com/zhangbingliang2019/DAPS`) all have publicly available code. We also make our code available at `https://github.com/Anima-Lab/DiffStateGrad`.

## 7 ACKNOWLEDGMENTS

R.Z. did the project under the Summer Undergraduate Research Fellowships (SURF) program at Caltech. R.Z. thanks the SURF program for funding this project. B.T. was supported by the Swartz Foundation Fellowship for Postdoctoral Research in Theoretical Neuroscience at Caltech. A.A. was supported by Bren Chair Professor and AI2050 Senior Fellow.

B.T. proposed and designed the project. R.Z. implemented the method and executed all the experiments, excluding the MRI experiment. B.T. ran the MRI experiment. R.Z and B.T. wrote the paper. A.A. supervised the project, provided valuable feedback, and offered editorial comments. R.Z. and B.T. prepared the rebuttal together. B.T. led the discussion during the review process.

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

# A  ADDITIONAL RESULTS

Table 5: **Robustness comparison of PSLD and DiffStateGrad-PSLD on linear tasks under different MG step sizes on FFHQ 256 × 256.**

| Method | Inpaint (Box) | | Inpaint (Random) | | Gaussian deblur | | Motion deblur | | SR (×4) | |
|---|---|---|---|---|---|---|---|---|---|---|
| | LPIPS↓ | PSNR↑ | LPIPS↓ | PSNR↑ | LPIPS↓ | PSNR↑ | LPIPS↓ | PSNR↑ | LPIPS↓ | PSNR↑ |
| *Default MG step size* | | | | | | | | | | |
| PSLD | 0.158 | 24.22 | 0.246 | 29.05 | 0.357 | 22.87 | 0.322 | 24.25 | **0.313** | 24.51 |
| DiffStateGrad-PSLD (ours) | 0.095 | 23.76 | 0.265 | 28.14 | 0.366 | 22.24 | 0.335 | 23.34 | 0.392 | 22.12 |
| *Large MG step size* | | | | | | | | | | |
| PSLD | 0.252 | 11.99 | 0.463 | 20.62 | 0.549 | 17.47 | 0.514 | 18.81 | 0.697 | 7.700 |
| DiffStateGrad-PSLD (ours) | **0.092** | **24.32** | **0.165** | **31.68** | **0.355** | **22.95** | **0.319** | **24.31** | 0.320 | **24.56** |

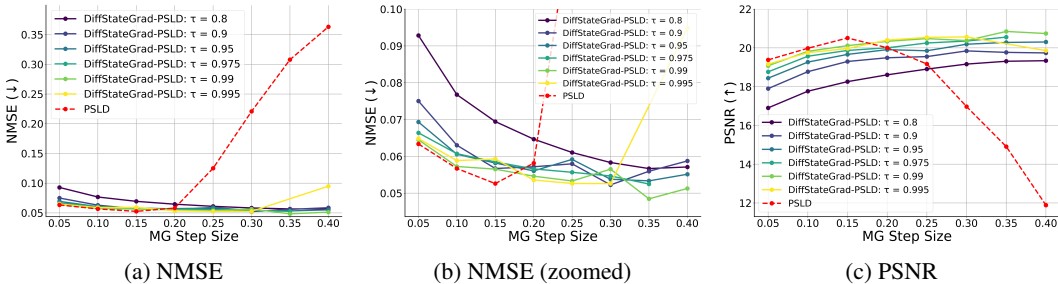

(a) NMSE      (b) NMSE (zoomed)      (c) PSNR

Figure 8: **Robustness comparison of PSLD and DiffStateGrad-PSLD for different variance retention thresholds** $\tau$. We evaluate images from FFHQ 256 × 256 on box inpainting.

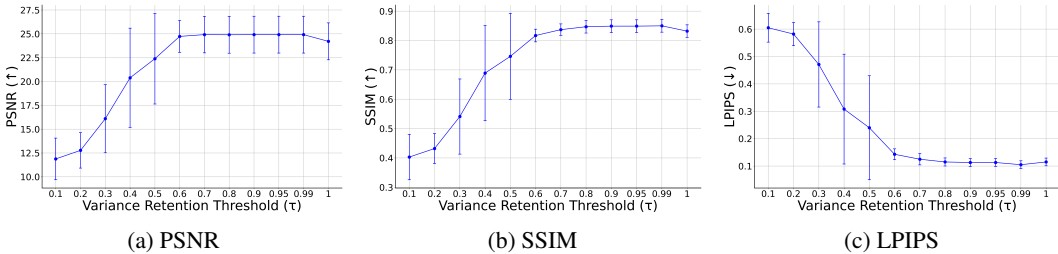

(a) PSNR      (b) SSIM      (c) LPIPS

Figure 9: **Performance of DiffStateGrad-DAPS for different variance retention thresholds** $\tau$. DiffStateGrad is robust to the choice of $\tau$, as values $\geq 0.6$ show similar performance in this figure. In our main experiments, we use $\tau = 0.99$ (see C.1 for further details). We evaluate images from FFHQ 256 × 256 on box inpainting.

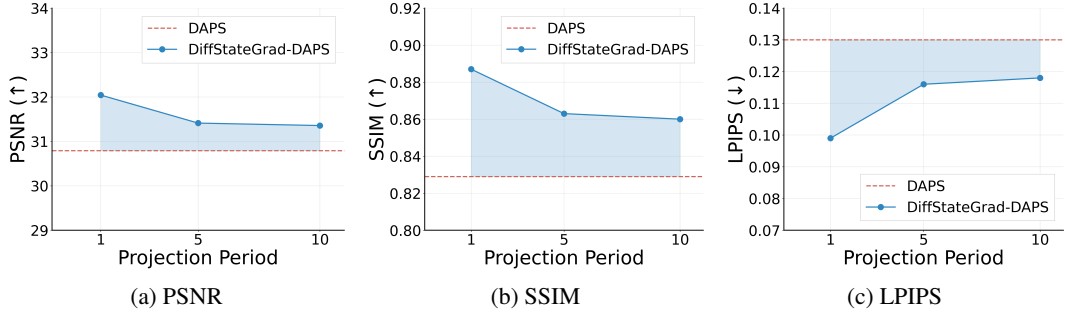

| (a) PSNR | (b) SSIM | (c) LPIPS |

Figure 10: **Performance of DiffStateGrad-DAPS for different projection periods** $P$. As period of projection increases, DiffStateGrad still outperforms DAPS without projection, which is the SOTA. See C.1 for details of $P$ values used in our experiments. We evaluate images from FFHQ $256 \times 256$ on random inpainting.

Table 6: **SSIM comparison on FFHQ $256 \times 256$.**

(a) Linear tasks.

| Method | Inpaint (Box) | Inpaint (Random) | Gaussian deblur | Motion deblur | SR (x4) |
|---|---|---|---|---|---|
| | SSIM↑ | SSIM↑ | SSIM↑ | SSIM↑ | SSIM↑ |
| *Pixel-based* | | | | | |
| DAPS | 0.806 (0.028) | 0.829 (0.022) | 0.786 (0.051) | 0.837 (0.040) | 0.797 (0.044) |
| DiffStateGrad-DAPS (ours) | **0.849** (0.029) | **0.887** (0.023) | **0.803** (0.044) | **0.853** (0.028) | **0.801** (0.039) |
| DPS | 0.810 (0.039) | 0.815 (0.045) | 0.709 (0.062) | 0.754 (0.056) | 0.675 (0.071) |
| DiffStateGrad-DPS (ours) | 0.831 (0.039) | 0.852 (0.046) | 0.739 (0.062) | 0.782 (0.056) | 0.683 (0.073) |
| MPGD-AE | 0.753 (0.029) | 0.731 (0.050) | 0.664 (0.071) | 0.723 (0.061) | 0.670 (0.070) |
| *Latent* | | | | | |
| PSLD | 0.819 (0.031) | 0.809 (0.049) | 0.537 (0.094) | 0.615 (0.075) | 0.650 (0.140) |
| DiffStateGrad-PSLD (ours) | **0.880** (0.028) | 0.898 (0.024) | 0.542 (0.077) | 0.620 (0.065) | 0.640 (0.123) |
| ReSample | 0.807 (0.036) | 0.892 (0.030) | 0.757 (0.049) | 0.854 (0.034) | 0.790 (0.048) |
| DiffStateGrad-ReSample (ours) | 0.841 (0.032) | **0.913** (0.023) | **0.767** (0.041) | **0.860** (0.031) | **0.795** (0.044) |

(b) Nonlinear tasks.

| Method | Phase retrieval | Nonlinear deblur | High dynamic range |
|---|---|---|---|
| | SSIM↑ | SSIM↑ | SSIM↑ |
| *Pixel-based* | | | |
| DAPS | 0.823 (0.033) | 0.723 (0.034) | 0.817 (0.109) |
| DiffStateGrad-DAPS (ours) | **0.868** (0.026) | **0.818** (0.035) | **0.852** (0.098) |
| *Latent* | | | |
| ReSample | 0.750 (0.246) | 0.842 (0.038) | 0.819 (0.109) |
| DiffStateGrad-ReSample (ours) | **0.855** (0.130) | **0.847** (0.035) | **0.857** (0.059) |

# B VISUALIZATIONS

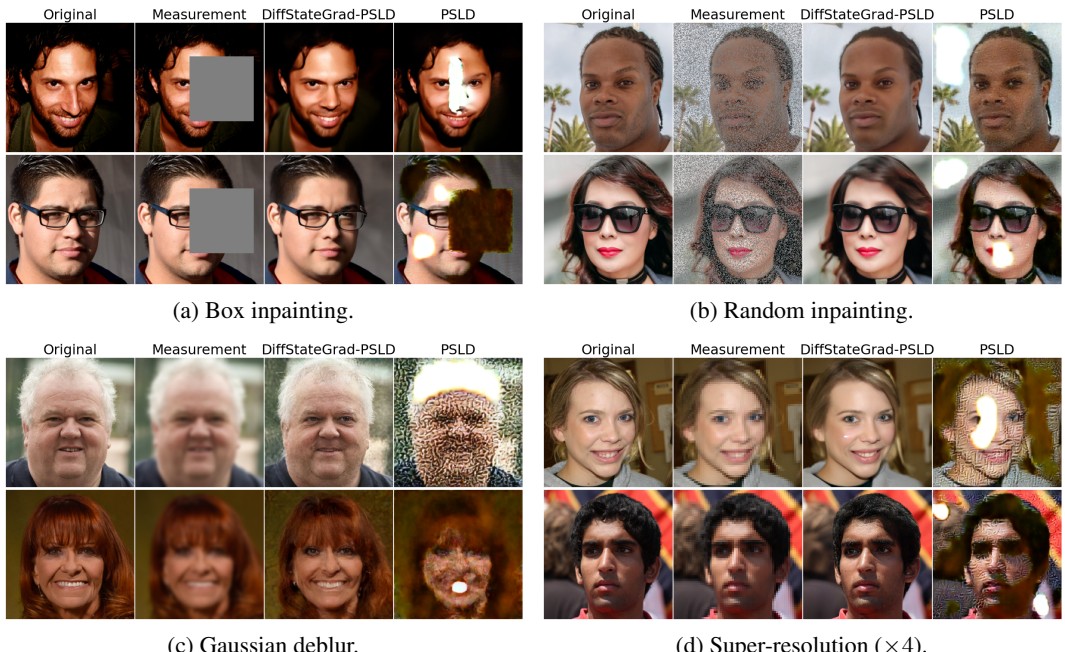

(a) Box inpainting.

(b) Random inpainting.

(c) Gaussian deblur.

(d) Super-resolution ($\times 4$).

Figure 11: **Qualitative comparison of DiffStateGrad-PSLD and PSLD for a large MG step size**. Images are chosen at random for visualization.

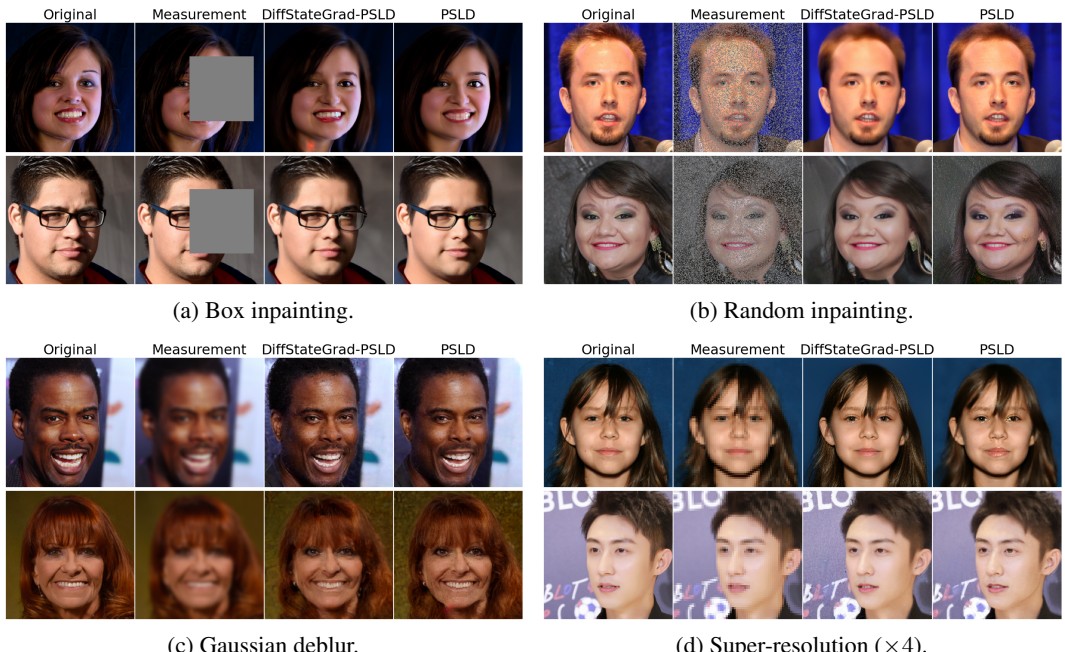

(a) Box inpainting.

(b) Random inpainting.

(c) Gaussian deblur.

(d) Super-resolution ($\times 4$).

Figure 12: **Qualitative comparison of DiffStateGrad-PSLD and PSLD for their best-performing MG step size**. Images are chosen at random for visualization.

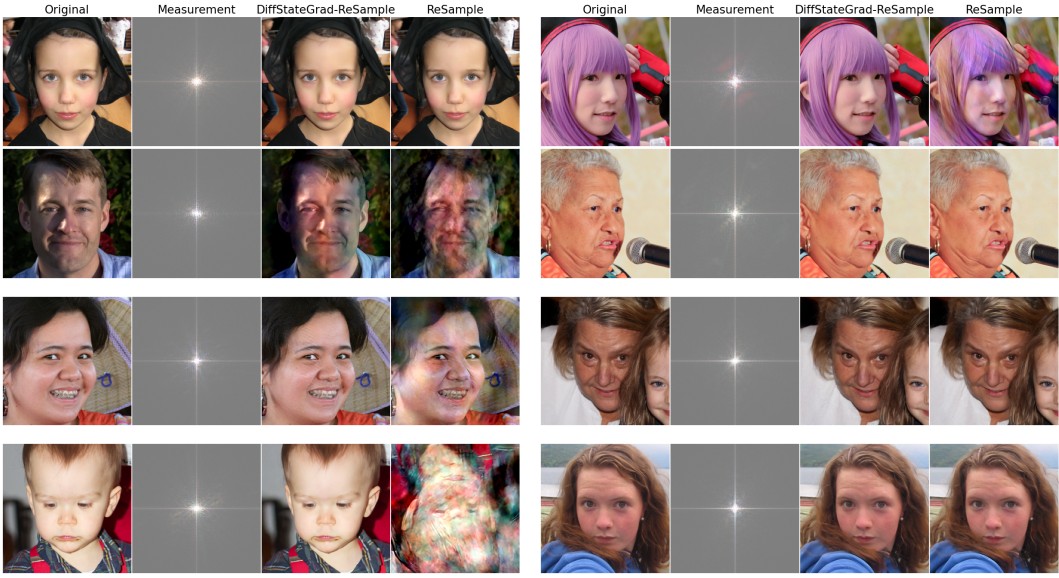

Figure 13: **Qualitative comparison of DiffStateGrad-ReSample and ReSample for phase retrieval**. Whereas the performance of ReSample is inconsistent, DiffStateGrad-ReSample consistently produces accurate reconstructions. Images are chosen at random for visualization.

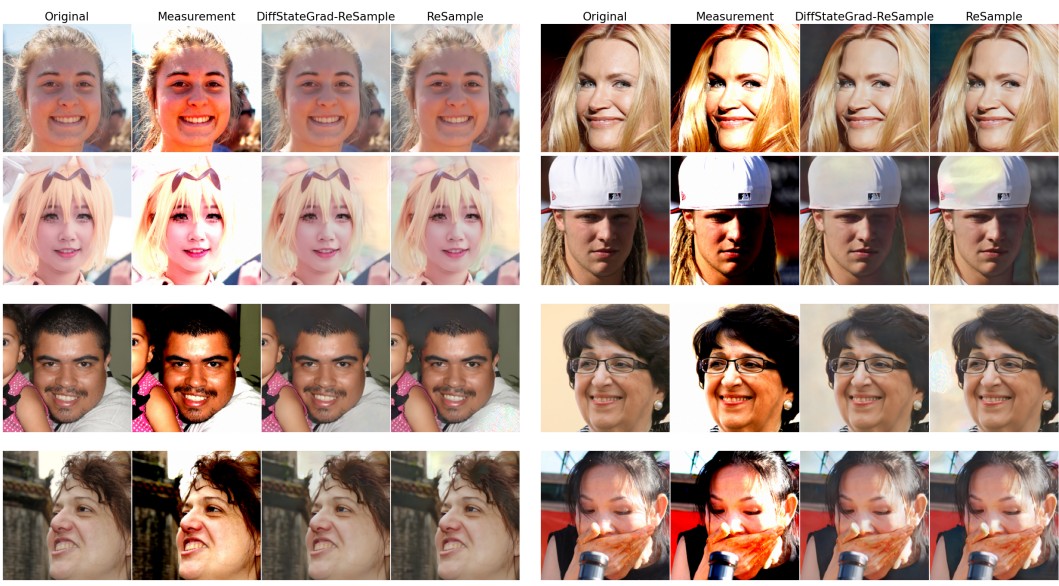

Figure 14: **Qualitative comparison of DiffStateGrad-ReSample and ReSample for high dynamic range (HDR)**. Images are chosen at random for visualization.

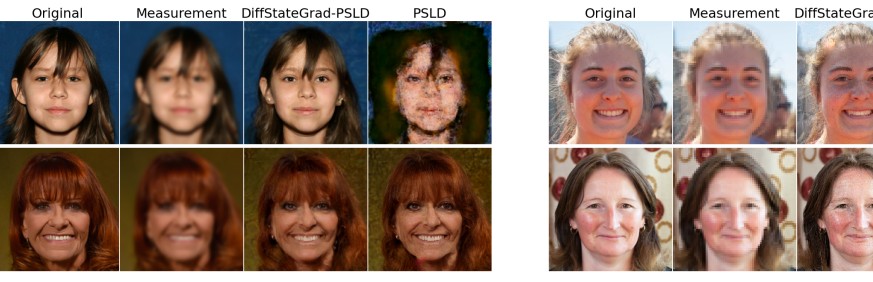

(a) Gaussian deblur.                                        (b) Super-resolution ($\times 4$).

Figure 15: **Qualitative comparison of DiffStateGrad-PSLD and PSLD for their best-performing MG step size**. DiffStateGrad-PSLD can remove artifacts and reduce failure cases, producing more reliable reconstructions. Images are chosen at random for visualization.

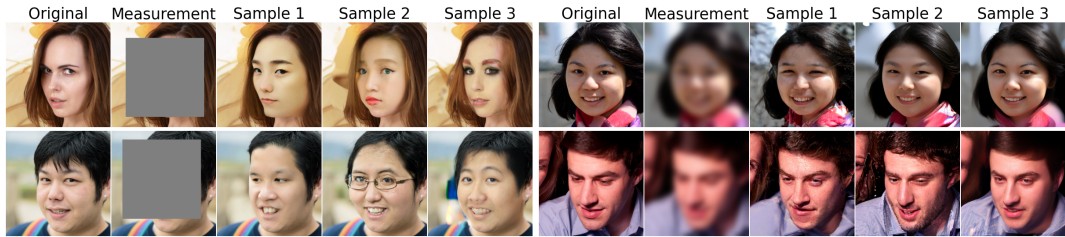

(a) Box inpainting ($192 \times 192$ box).              (b) Gaussian deblur ($81 \times 81$ kernel, SD of 7).

Figure 16: **Reconstruction diversity of DiffStateGrad.** DiffStateGrad-PSLD with a large MG step size can produce a diverse range of images from multimodal posteriors. Generated images have distinctive facial features.

## C    IMPLEMENTATION DETAILS

### C.1    HYPERPARAMETERS

For all main experiments across all four methods, we use the variance retention threshold $\tau = 0.99$. For all experiments involving PSLD, DPS, and DAPS, we perform the DiffStateGrad projection step every iteration ($P = 1$). For all experiments involving ReSample, we perform the step every five iterations ($P = 5$). See the sections dedicated to each method for further implementation details. We reiterate that various values of $\tau$ and $P$ are reasonable options for optimal performance (see Figures 8 to 10).

### C.2    EFFICIENCY EXPERIMENT

We evaluate the computational overhead introduced by DiffStateGrad across three diffusion-based methods: PSLD, ReSample, and DAPS. We conduct these experiments on the box inpainting task using an NVIDIA GeForce RTX 4090 GPU with 24GB of VRAM. Each method is run with its default settings on a set of 100 images from FFHQ $256 \times 256$, and we measure the average runtime in seconds per image.

### C.3    PSLD

Our DiffStateGrad-PSLD algorithm integrates the state-guided projected gradient directly into the PSLD update process. For each iteration (i.e., period $P = 1$) of the main loop, after computing the standard PSLD update $z'_{t-1}$, we introduce our DiffStateGrad method. First, we calculate the full gradient $G_t$ according to PSLD, combining both the measurement consistency term and the fixed-point constraint. We then perform SVD on the current latent representation (or diffusion state)

$\boldsymbol{Z}_t$ ($\boldsymbol{z}_t$ in image matrix form). Using the variance retention threshold $\tau$, we determine the appropriate rank for our projection. We construct projection matrices from the truncated singular vectors and use these to approximate the gradient. This approximated gradient $\boldsymbol{G}'_t$ is then used for the final update step, replacing the separate gradient updates in standard PSLD. This process is repeated at every iteration, allowing for adaptive, low-rank updates throughout the entire diffusion process.

For experiments, we use the official implementation of PSLD (Rout et al., 2023) with default configurations (i.e., noise, step size, etc.) for reproducing baselines.

---

**Algorithm 2** DiffStateGrad-PSLD for Image Restoration Tasks

---

**Require:** $T, \boldsymbol{y}, \{\eta_t\}_{t=1}^T, \{\gamma_t\}_{t=1}^T, \{\tilde{\sigma}_t\}_{t=1}^T$
**Require:** $\mathcal{E}, \mathcal{D}, \mathcal{A}\boldsymbol{x}_0^*, \mathcal{A}, \boldsymbol{s}_\theta,$ variance retention threshold $\tau$
1: $\boldsymbol{z}_T \sim \mathcal{N}(\boldsymbol{0}, \boldsymbol{I})$
2: **for** $t = T - 1$ **to** $0$ **do**
3: $\quad \hat{\boldsymbol{s}} \leftarrow \boldsymbol{s}_\theta(\boldsymbol{z}_t, t)$
4: $\quad \hat{\boldsymbol{z}}_0 \leftarrow \frac{1}{\sqrt{\bar{\alpha}_t}}(\boldsymbol{z}_t + (1 - \bar{\alpha}_t)\hat{\boldsymbol{s}})$
5: $\quad \boldsymbol{\epsilon} \sim \mathcal{N}(\boldsymbol{0}, \boldsymbol{I})$
6: $\quad \boldsymbol{z}'_{t-1} \leftarrow \frac{\sqrt{\alpha_t}(1 - \bar{\alpha}_{t-1})}{1 - \bar{\alpha}_t}\boldsymbol{z}_t + \frac{\sqrt{\bar{\alpha}_{t-1}}\beta_t}{1 - \bar{\alpha}_t}\hat{\boldsymbol{z}}_0 + \tilde{\sigma}_t\boldsymbol{\epsilon}$
7: $\quad \boldsymbol{g}_t \leftarrow \eta_t\nabla_{\boldsymbol{z}_t}\|\boldsymbol{y} - \mathcal{A}(\mathcal{D}(\hat{\boldsymbol{z}}_0))\|_2^2 + \gamma_t\nabla_{\boldsymbol{z}_t}\|\hat{\boldsymbol{z}}_0 - \mathcal{E}(\mathcal{A}^T\mathcal{A}\boldsymbol{x}_0^* + (\boldsymbol{I} - \mathcal{A}^T\mathcal{A})\mathcal{D}(\hat{\boldsymbol{z}}_0))\|_2^2$
8: $\quad \boldsymbol{U}, \boldsymbol{S}, \boldsymbol{V} \leftarrow \text{SVD}(\boldsymbol{Z}_t)$
9: $\quad \lambda_j \leftarrow s_j^2$ ($s_j$ are the singular values of $\boldsymbol{S}$)
10: $\quad c_k \leftarrow \frac{\sum_{j=1}^k \lambda_j}{\sum_j \lambda_j}$
11: $\quad r \leftarrow \arg\min_k\{c_k \geq \tau\}$
12: $\quad \boldsymbol{A}_t \leftarrow \boldsymbol{U}_r$
13: $\quad \boldsymbol{B}_t \leftarrow \boldsymbol{V}_r$
14: $\quad \boldsymbol{R}_t \leftarrow \boldsymbol{A}_t^T\boldsymbol{G}_t\boldsymbol{B}_t^T$
15: $\quad \boldsymbol{G}'_t \leftarrow \boldsymbol{A}_t\boldsymbol{R}_t\boldsymbol{B}_t$
16: $\quad \boldsymbol{z}_{t-1} \leftarrow \boldsymbol{z}'_{t-1} - \boldsymbol{g}'_t$
17: **end for**
18: **return** $\mathcal{D}(\hat{\boldsymbol{z}}_0)$

---

## C.4 RESAMPLE

Our DiffStateGrad-ReSample algorithm integrates the state-guided projected gradient into the optimization process of ReSample (Song et al., 2023a). We introduce two new hyperparameters: the variance retention threshold $\tau$ and a period $P$ for applying our DiffStateGrad step. During each ReSample step, we first perform SVD on the current latent representation (or diffusion state) $\boldsymbol{Z}'_t$ ($\boldsymbol{z}'_t$ in image matrix form). Note that we do not perform SVD within the gradient descent loop itself, meaning that we only perform SVD at most once per iteration of the sampling algorithm. We then determine the appropriate rank based on $\tau$ and construct projection matrices. Then, within the gradient descent loop for solving $\hat{\boldsymbol{z}}_0(\boldsymbol{y})$, we approximate the gradient in the diffusion state subspace using our projection matrices every $P = 5$ steps. On steps where DiffStateGrad is not applied, we use the standard gradient. This adaptive, periodic application of DiffStateGrad allows for a balance between the benefits of low-rank approximation and the potential need for full gradient information. The rest of the ReSample algorithm, including the stochastic resampling step, remains unchanged.

We note that the ReSample algorithm employs a two-stage approach for its hard data consistency step. Initially, it performs pixel-space optimization. This step is computationally efficient and produces smoother, albeit potentially blurrier, results with high-level semantic information. As the diffusion process approaches $t = 0$, ReSample transitions to latent-space optimization to refine the image with finer details. Our DiffStateGrad method is specifically integrated into this latter, latent-space optimization stage. By applying DiffStateGrad to the latent optimization, we aim to mitigate the potential introduction of artifacts and off-manifold deviations that can occur due to the direct manipulation of latent variables. This application of DiffStateGrad allows us to benefit from the computational efficiency of initial pixel-space optimization while enhancing the robustness and

quality of the final latent-space refinement. Importantly, DiffStateGrad is not applied during the pixel-space optimization phase, as this stage already tends to produce smoother results and is less prone to artifact introduction.

For experiments, we use the official implementation of ReSample (Song et al., 2023a) with default configurations (i.e., noise, step size, etc.) for reproducing baselines.

---

**Algorithm 3** DiffStateGrad-ReSample for Image Restoration Tasks

---

**Require:** Measurements $\boldsymbol{y}$, $\mathcal{A}(\cdot)$, Encoder $\mathcal{E}(\cdot)$, Decoder $\mathcal{D}(\cdot)$, Score function $\boldsymbol{s}_\theta(\cdot, t)$, Pretrained LDM Parameters $\beta_t$, $\bar{\alpha}_t$, $\eta$, $\delta$, Hyperparameter $\gamma$ to control $\sigma_t^2$, Time steps to perform resample $C$, Variance retention threshold $\tau$, Period $P$

1: $\boldsymbol{z}_T \sim \mathcal{N}(\boldsymbol{0}, \boldsymbol{I})$                                    ▷ Initial noise vector
2: **for** $t = T - 1, \ldots, 0$ **do**
3:     $\boldsymbol{\epsilon}_1 \sim \mathcal{N}(\boldsymbol{0}, \boldsymbol{I})$
4:     $\hat{\boldsymbol{\epsilon}}_{t+1} = \boldsymbol{s}_\theta(\boldsymbol{z}_{t+1}, t+1)$                        ▷ Compute the score
5:     $\hat{\boldsymbol{z}}_0(\boldsymbol{z}_{t+1}) = \frac{1}{\sqrt{\bar{\alpha}_{t+1}}}(\boldsymbol{z}_{t+1} - \sqrt{1 - \bar{\alpha}_{t+1}}\hat{\boldsymbol{\epsilon}}_{t+1})$    ▷ Predict $\hat{\boldsymbol{z}}_0$ using Tweedie's formula
6:     $\boldsymbol{z}_t' = \sqrt{\bar{\alpha}_t}\hat{\boldsymbol{z}}_0(\boldsymbol{z}_{t+1}) + \sqrt{1 - \bar{\alpha}_t - \eta\delta^2}\hat{\boldsymbol{\epsilon}}_{t+1} + \eta\delta\boldsymbol{\epsilon}_1$    ▷ Unconditional DDIM step
7:     **if** $t \in C$ **then**                                      ▷ ReSample time step
8:         Initialize $\hat{\boldsymbol{z}}_0(\boldsymbol{y})$ with $\hat{\boldsymbol{z}}_0(\boldsymbol{z}_{t+1})$
9:         $\boldsymbol{U}, \boldsymbol{S}, \boldsymbol{V} \leftarrow \text{SVD}(\boldsymbol{Z}_t')$                   ▷ Perform SVD on current diffusion state
10:        $\lambda_j \leftarrow s_j^2$ (where $s_j$ are the singular values of $\boldsymbol{S}$)    ▷ Calculate eigenvalues
11:        $c_k \leftarrow \frac{\sum_{j=1}^k \lambda_j}{\sum_j \lambda_j}$                          ▷ Cumulative sum of eigenvalues
12:        $r \leftarrow \arg\min_k\{c_k \geq \tau\}$                   ▷ Determine rank $r$ based on threshold $\tau$
13:        $\boldsymbol{A} \leftarrow \boldsymbol{U}_r$                                     ▷ Get first $r$ left singular vectors
14:        $\boldsymbol{B} \leftarrow \boldsymbol{V}_r$                                     ▷ Get first $r$ right singular vectors
15:        **for** each step in gradient descent **do**
16:            **if** step number $\mod P = 0$ **then**
17:                $\boldsymbol{g} \leftarrow \nabla_{\hat{\boldsymbol{z}}_0(\boldsymbol{y})}\frac{1}{2}\|\boldsymbol{y} - \mathcal{A}(\mathcal{D}(\hat{\boldsymbol{z}}_0(\boldsymbol{y})))\|_2^2$    ▷ Compute gradient
18:                $\boldsymbol{R} \leftarrow \boldsymbol{A}^T \boldsymbol{G} \boldsymbol{B}^T$                        ▷ Project gradient
19:                $\boldsymbol{G}' \leftarrow \boldsymbol{A} \boldsymbol{R} \boldsymbol{B}$                   ▷ Reconstruct approximated gradient
20:            **else**
21:                $\boldsymbol{g}' \leftarrow \nabla_{\hat{\boldsymbol{z}}_0(\boldsymbol{y})}\frac{1}{2}\|\boldsymbol{y} - \mathcal{A}(\mathcal{D}(\hat{\boldsymbol{z}}_0(\boldsymbol{y})))\|_2^2$    ▷ Compute gradient without modification
22:            **end if**
23:            Update $\hat{\boldsymbol{z}}_0(\boldsymbol{y})$ using gradient $\boldsymbol{g}'$
24:        **end for**
25:        $\boldsymbol{z}_t = \text{StochasticResample}(\hat{\boldsymbol{z}}_0(\boldsymbol{y}), \boldsymbol{z}_t', \gamma)$                ▷ Map back to $t$
26:    **else**
27:        $\boldsymbol{z}_t = \boldsymbol{z}_t'$                               ▷ Unconditional sampling if not resampling
28:    **end if**
29: **end for**
30: $\boldsymbol{x}_0 = \mathcal{D}(\boldsymbol{z}_0)$                                        ▷ Output reconstructed image
31: **return** $\boldsymbol{x}_0$

---

## C.5   DPS

Like DiffStateGrad-PSLD, our DiffStateGrad-DPS algorithm integrates the state-guided projected gradient into each iteration of the DPS process. The key difference lies in operating directly on pixel-based diffusion states rather than latent representations. After computing the standard DPS update $\boldsymbol{x}_{t-1}'$, we follow the same projection strategy: performing SVD on the current diffusion state $\boldsymbol{X}_t$, determining projection rank using threshold $\tau$, and constructing an approximated gradient $\boldsymbol{G}_t'$ that replaces the standard DPS gradient update.

For experiments, we use the official implementation of DPS (Chung et al., 2023) with default configurations (i.e., noise, step size, etc.) for reproducing baselines.

---

**Algorithm 4** DiffStateGrad-DPS for Image Restoration Tasks

---

**Require:** $T, \boldsymbol{y}, \{\zeta_t\}_{t=1}^T, \{\tilde{\sigma}_t\}_{t=1}^T, \boldsymbol{s}_\theta,$ variance retention threshold $\tau$
 1: $\boldsymbol{x}_T \sim \mathcal{N}(\boldsymbol{0}, \boldsymbol{I})$
 2: **for** $t = T - 1$ **to** $0$ **do**
 3: $\quad \hat{\boldsymbol{s}} \leftarrow \boldsymbol{s}_\theta(\boldsymbol{x}_t, t)$
 4: $\quad \hat{\boldsymbol{x}}_0 \leftarrow \frac{1}{\sqrt{\bar{\alpha}_t}}(\boldsymbol{x}_t + (1 - \bar{\alpha}_t)\hat{\boldsymbol{s}})$
 5: $\quad \boldsymbol{z} \sim \mathcal{N}(\boldsymbol{0}, \boldsymbol{I})$
 6: $\quad \boldsymbol{x}'_{t-1} \leftarrow \frac{\sqrt{\alpha_t}(1-\bar{\alpha}_{t-1})}{1-\bar{\alpha}_t}\boldsymbol{x}_t + \frac{\sqrt{\bar{\alpha}_{t-1}}\beta_t}{1-\bar{\alpha}_t}\hat{\boldsymbol{x}}_0 + \tilde{\sigma}_t\boldsymbol{z}$
 7: $\quad \boldsymbol{g}_t \leftarrow \zeta_t \nabla_{\boldsymbol{x}_t}\|\boldsymbol{y} - \mathcal{A}(\hat{\boldsymbol{x}}_0)\|_2^2$
 8: $\quad \boldsymbol{U}, \boldsymbol{S}, \boldsymbol{V} \leftarrow \text{SVD}(\boldsymbol{X}_t)$
 9: $\quad \lambda_j \leftarrow s_j^2$ ($s_j$ are the singular values of $\boldsymbol{S}$)
10: $\quad c_k \leftarrow \frac{\sum_{j=1}^k \lambda_j}{\sum_j \lambda_j}$
11: $\quad r \leftarrow \arg\min_k \{c_k \geq \tau\}$
12: $\quad \boldsymbol{A}_t \leftarrow \boldsymbol{U}_r$
13: $\quad \boldsymbol{B}_t \leftarrow \boldsymbol{V}_r$
14: $\quad \boldsymbol{R}_t \leftarrow \boldsymbol{A}_t^T \boldsymbol{G}_t \boldsymbol{B}_t^T$
15: $\quad \boldsymbol{G}'_t \leftarrow \boldsymbol{A}_t \boldsymbol{R}_t \boldsymbol{B}_t$
16: $\quad \boldsymbol{x}_{t-1} \leftarrow \boldsymbol{x}'_{t-1} - \boldsymbol{g}'_t$
17: **end for**
18: **return** $\hat{\boldsymbol{x}}_0$

---

## C.6 DAPS

We improve upon DAPS by incorporating a state-guided projected gradient. We introduce a variance retention threshold $\tau$ to determine the projection rank. For each noise level in the annealing loop, DAPS computes the initial estimate $\hat{\boldsymbol{x}}_0^{(0)}$ by solving the probability flow ODE using the score model $\boldsymbol{s}_\theta$. This estimate represents a guess of the clean image given the current noisy sample $\boldsymbol{x}_{t_i}$. We then perform SVD on this estimate in image matrix form $\hat{\boldsymbol{X}}_0^{(0)}$ (using it as our diffusion state), determine the appropriate rank based on $\tau$, and construct projection matrices. Within the Langevin dynamics loop, we calculate the full gradient, combining both the prior term $\log p(\hat{\boldsymbol{x}}_0^{(j)}|\boldsymbol{x}_{t_i})$ and the likelihood term $\log p(\boldsymbol{y}|\hat{\boldsymbol{x}}_0^{(j)})$. For each step (i.e., period $P = 1$), we project this gradient using our pre-computed matrices and use this projected gradient for the update step. This process is repeated for each noise level, progressively refining our estimate of the clean image.

For experiments, we use the official implementation of DAPS (Zhang et al., 2024) with default configurations (i.e., noise, step size, etc.) for reproducing baselines.

---

**Algorithm 5** DiffStateGrad-DAPS for Image Restoration Tasks

---

**Require:** Score model $s_\theta$, measurement $y$, noise schedule $\sigma_t$, $\{t_i\}_{i \in \{0,\ldots,N_A\}}$, variance retention threshold $\tau$
1: Sample $\boldsymbol{x}_T \sim \mathcal{N}(\boldsymbol{0}, \sigma_T^2 \boldsymbol{I})$
2: **for** $i = N_A, N_A - 1, \ldots, 1$ **do**
3:     Compute $\hat{\boldsymbol{x}}_0^{(0)} = \hat{\boldsymbol{x}}_0(\boldsymbol{x}_{t_i})$ by solving the probability flow ODE in Eq. (39) with $s_\theta$
4:     $\boldsymbol{U}, \boldsymbol{S}, \boldsymbol{V} \leftarrow \text{SVD}(\hat{\boldsymbol{X}}_0^{(0)})$                                ▷ Perform SVD on initial estimate
5:     $\lambda_j \leftarrow s_j^2$ (where $s_j$ are the singular values of $\boldsymbol{S}$)              ▷ Calculate eigenvalues
6:     $c_k \leftarrow \frac{\sum_{j=1}^k \lambda_j}{\sum_j \lambda_j}$                                    ▷ Cumulative sum of eigenvalues
7:     $r \leftarrow \arg\min_k \{c_k \geq \tau\}$                    ▷ Determine rank $r$ based on threshold $\tau$
8:     $\boldsymbol{A} \leftarrow \boldsymbol{U}_r$                                    ▷ Get first $r$ left singular vectors
9:     $\boldsymbol{B} \leftarrow \boldsymbol{V}_r$                                    ▷ Get first $r$ right singular vectors
10:     **for** $j = 0, \ldots, N - 1$ **do**
11:         $\boldsymbol{g} \leftarrow \nabla_{\hat{\boldsymbol{x}}_0} \log p(\hat{\boldsymbol{x}}_0^{(j)} | \boldsymbol{x}_{t_i}) + \nabla_{\hat{\boldsymbol{x}}_0} \log p(\boldsymbol{y} | \hat{\boldsymbol{x}}_0^{(j)})$            ▷ Compute gradient
12:         $\boldsymbol{R} \leftarrow \boldsymbol{A}^T \boldsymbol{G} \boldsymbol{B}^T$                                  ▷ Project gradient
13:         $\boldsymbol{G}' \leftarrow \boldsymbol{A} \boldsymbol{R} \boldsymbol{B}$                   ▷ Reconstruct approximated gradient
14:         $\hat{\boldsymbol{x}}_0^{(j+1)} \leftarrow \hat{\boldsymbol{x}}_0^{(j)} + \eta_t \boldsymbol{g}' + \sqrt{2\eta_t} \boldsymbol{\epsilon}_j, \boldsymbol{\epsilon}_j \sim \mathcal{N}(\boldsymbol{0}, \boldsymbol{I})$
15:     **end for**
16:     Sample $\boldsymbol{x}_{t_{i-1}} \sim \mathcal{N}(\hat{\boldsymbol{x}}_0^{(N)}, \sigma_{t_{i-1}}^2 \boldsymbol{I})$
17: **end for**
18: **return** $\boldsymbol{x}_0$

---

### C.7 DDNM

We report numbers from DAPS (Zhang et al., 2024), which conducts experiments under identical settings to ours.

### C.8 DDRM

We report numbers from DAPS (Zhang et al., 2024), which conducts experiments under identical settings to ours.

### C.9 MCG

We report numbers from DPS (Chung et al., 2023), which conducts experiments under identical settings to ours.

### C.10 MPGD-AE

For experiments, we use the official implementation of MPGD-AE (He et al., 2024). For accurate comparison, we use 1000 DDIM steps and the guidance weight parameter of 0.5 (which we find through fine-tuning).

## D PROOFS

**Proposition 1.** *Let $\mathcal{M}$ be a smooth $m$-dimensional submanifold of a $d$-dimensional Euclidean space $\mathbb{R}^d$, where $m < d$. Assume that for each state $\boldsymbol{z}_t \in \mathcal{M}$, the tangent space $T_{\boldsymbol{z}_t}\mathcal{M}$ is well-defined, and the projection operator $\mathcal{P}_{\mathcal{S}_{\boldsymbol{z}_t}}$ onto an approximate subspace $\mathcal{S}_{\boldsymbol{z}_t}$ closely approximates the projection onto $T_{\boldsymbol{z}_t}\mathcal{M}$. For the state $\boldsymbol{z}_t \in \mathcal{M}$ and measurement gradient $\boldsymbol{g}_t \in \mathbb{R}^d$, consider two update rules:*

$$\begin{aligned} \boldsymbol{z}_{t-1} &= \boldsymbol{z}_t - \eta \boldsymbol{g}_t \quad \textit{(standard update)}, \\ \boldsymbol{z}'_{t-1} &= \boldsymbol{z}_t - \eta \mathcal{P}_{\mathcal{S}_{\boldsymbol{z}_t}}(\boldsymbol{g}_t) \quad \textit{(projected update)}, \end{aligned} \tag{7}$$

*where $\eta > 0$ is a small step size. Then, for sufficiently small $\eta$, the projected update $z'_{t-1}$ stays closer to the manifold $\mathcal{M}$ than the standard update $z_{t-1}$. That is,*

$$dist(z'_{t-1}, \mathcal{M}) < dist(z_{t-1}, \mathcal{M}). \tag{8}$$

*Proof.* Let $z_t \in \mathcal{M}$ and $g_t \in \mathbb{R}^d$. Decompose the gradient $g_t$ into components tangent and normal to $\mathcal{M}$ at $z_t$:

$$g_t = g_t^{\|} + g_t^{\perp},$$

where $g_t^{\|} \in T_{z_t}\mathcal{M}$ and $g_t^{\perp} \in N_{z_t}\mathcal{M}$, the normal space at $z_t$.

We have two projection operators:

- $\mathcal{P}_{T_{z_t}\mathcal{M}}$: the exact orthogonal projection onto the tangent space $T_{z_t}\mathcal{M}$.

- $\mathcal{P}_{\mathcal{S}_{z_t}}$: an approximate projection operator onto a subspace $\mathcal{S}_{z_t}$ that closely approximates $T_{z_t}\mathcal{M}$.

Assuming that $\mathcal{P}_{\mathcal{S}_{z_t}}$ approximates $\mathcal{P}_{T_{z_t}\mathcal{M}}$, we have:

$$\mathcal{P}_{\mathcal{S}_{z_t}}(g_t) = g_t^{\|} + \epsilon,$$

where $\epsilon = \mathcal{P}_{\mathcal{S}_{z_t}}(g_t) - g_t^{\|}$ is the approximation error, which is small.

The standard update is:

$$z_{t-1} = z_t - \eta g_t = z_t - \eta(g_t^{\|} + g_t^{\perp}).$$

The projected update is:

$$z'_{t-1} = z_t - \eta \mathcal{P}_{\mathcal{S}_{z_t}}(g_t) = z_t - \eta(g_t^{\|} + \epsilon).$$

Let $\pi(z)$ denote the orthogonal projection of $z$ onto $\mathcal{M}$. For points close to $z_t$, we can approximate $\pi(z)$ using the tangent space projection, which comes from the first-order Taylor expansion of $\mathcal{M}$ at $z_t$:

$$\pi(z) \approx z_t + \mathcal{P}_{T_{z_t}\mathcal{M}}(z - z_t).$$

Here, the higher-order terms for $\pi(z_{t-1})$ are of order $\mathcal{O}(\|z_{t-1} - z_t\|^2) = \mathcal{O}((\eta\|g_t\|)^2) = \mathcal{O}(\eta^2)$. Similarly, the higher-order terms for $\pi(z'_{t-1})$ are of order $\mathcal{O}(\|z'_{t-1} - z_t\|^2) = \mathcal{O}((\eta\|\mathcal{P}_{\mathcal{S}_{z_t}}(g_t)\|)^2) = \mathcal{O}(\eta^2)$. We will address these terms at the end of the proof.

First, compute the distance from $z_{t-1}$ to $\mathcal{M}$:

$$\begin{aligned}
dist(z_{t-1}, \mathcal{M}) &= \|z_{t-1} - \pi(z_{t-1})\| \\
&\approx \left\| z_t - \eta(g_t^{\|} + g_t^{\perp}) - \left( z_t + \mathcal{P}_{T_{z_t}\mathcal{M}}(-\eta(g_t^{\|} + g_t^{\perp})) \right) \right\| \\
&= \left\| -\eta(g_t^{\|} + g_t^{\perp}) + \eta \mathcal{P}_{T_{z_t}\mathcal{M}}(g_t^{\|} + g_t^{\perp}) \right\| \\
&= \left\| -\eta g_t^{\perp} + \eta \mathcal{P}_{T_{z_t}\mathcal{M}}(g_t^{\perp}) \right\| \\
&= \eta \left\| g_t^{\perp} - \mathcal{P}_{T_{z_t}\mathcal{M}}(g_t^{\perp}) \right\|.
\end{aligned}$$

Since $g_t^{\perp} \in N_{z_t}\mathcal{M}$ and $\mathcal{P}_{T_{z_t}\mathcal{M}}(g_t^{\perp}) = 0$, we have:

$$dist(z_{t-1}, \mathcal{M}) = \eta \left\| g_t^{\perp} \right\|. \tag{9}$$

Now, compute the distance from $z'_{t-1}$ to $\mathcal{M}$:

$$
\begin{aligned}
\text{dist}(z'_{t-1}, \mathcal{M}) &= \left\| z'_{t-1} - \pi(z'_{t-1}) \right\| \\
&\approx \left\| z_t - \eta(g_t^\parallel + \epsilon) - \left( z_t + \mathcal{P}_{T_{z_t}\mathcal{M}}(-\eta(g_t^\parallel + \epsilon)) \right) \right\| \\
&= \left\| -\eta(g_t^\parallel + \epsilon) + \eta\mathcal{P}_{T_{z_t}\mathcal{M}}(g_t^\parallel + \epsilon) \right\| \\
&= \left\| -\eta\epsilon + \eta\mathcal{P}_{T_{z_t}\mathcal{M}}(\epsilon) \right\| \\
&= \eta \left\| \epsilon - \mathcal{P}_{T_{z_t}\mathcal{M}}(\epsilon) \right\| \\
&= \eta \left\| (I - \mathcal{P}_{T_{z_t}\mathcal{M}})\epsilon \right\|.
\end{aligned}
$$

Since $\left\| \epsilon^\perp \right\| = \left\| (I - \mathcal{P}_{T_{z_t}\mathcal{M}})\epsilon \right\|$, we have:

$$
\text{dist}(z'_{t-1}, \mathcal{M}) = \eta \left\| \epsilon^\perp \right\|. \tag{10}
$$

Because $\epsilon$ is small, we can bound $\left\| \epsilon^\perp \right\| \leq c' \left\| g_t^\perp \right\|$ for some small constant $c' > 0$.

Therefore,

$$
\text{dist}(z'_{t-1}, \mathcal{M}) \leq c'\eta \left\| g_t^\perp \right\|.
$$

Comparing the distances, we have:

$$
\begin{aligned}
\text{dist}(z_{t-1}, \mathcal{M}) - \text{dist}(z'_{t-1}, \mathcal{M}) &\geq \eta \left\| g_t^\perp \right\| - c'\eta \left\| g_t^\perp \right\| \\
&= (1 - c')\eta \left\| g_t^\perp \right\| \\
&= c\eta \left\| g_t^\perp \right\|,
\end{aligned}
$$

where $c = 1 - c' > 0$.

Including higher-order terms, we can write:

$$
\text{dist}(z'_{t-1}, \mathcal{M}) \leq \text{dist}(z_{t-1}, \mathcal{M}) - c\eta \left\| g_t^\perp \right\| + \mathcal{O}(\eta^2). \tag{11}
$$

Therefore, for sufficiently small $\eta$, the linear term dominates the higher-order terms, meaning that the projected update $z'_{t-1}$ stays closer to the manifold $\mathcal{M}$ than the standard update $z_{t-1}$:

$$
\text{dist}(z'_{t-1}, \mathcal{M}) < \text{dist}(z_{t-1}, \mathcal{M}).
$$

$\square$

## E  MRI EXPERIMENTS

To demonstrate the applicability of DiffStateGrad beyond natural images and the discussed diffusion-based inverse problems, we conduct an additional experiment on Magnetic Resonance Imaging (MRI); this represents an inverse problem where undersampled measurements are observed in the frequency domain, and the task is to reconstruct high-quality MRI in the image space. We utilize the Compressed Sensing Generative Model (Jalal et al., 2021), which employs Langevin dynamics for MRI reconstruction (available at https://github.com/utcsilab/csgm-mri-langevin). This method, which we refer to as CSGM-Langevin, is built upon prior works (Song & Ermon, 2020; 2019).

We incorporate our proposed DiffStateGrad into the measurement gradient guidance. Specifically, given the complex nature of MRI data, we apply DiffStateGrad to the magnitude of the complex-valued data while preserving the phase information. This framework uses $x_t$ for both measurement gradient computation and incorporation. The unconditional diffusion model was trained on T2-weighted brain datasets from the NYU fastMRI dataset (Zbontar et al., 2018; Knoll et al., 2020), and the reported results were averaged over 30 test examples (reporting avg (std)). The measurement operators give samples vertically equispaced in $k$-space at an undersampling rate of $R = 4$.

While the default optimized measurement gradient step size from (Jalal et al., 2021) is 5.0, the results confirm that DiffStateGrad enhances the robustness of CSGM-Langevin when the step size deviates

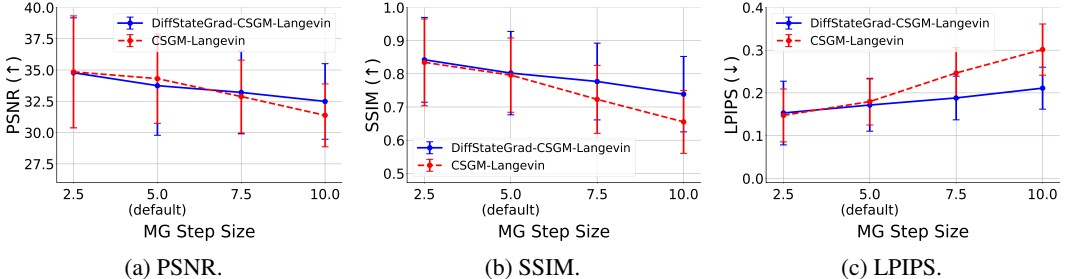

(a) PSNR.        (b) SSIM.        (c) LPIPS.

Figure 17: **Robustness of DiffStateGrad to MG step size in MRI**. Comparing CSGM-Langevin (Jalal et al., 2021) with DiffStateGrad-CSGM-Langevin on 30 examples from the test set of the T2-weighted brain fastMRI dataset (Zbontar et al., 2018; Knoll et al., 2020). The measurement operator gives vertically and equispaced samples in the $k$-space with the undersampling rate of $R = 4$.

from this optimal value. For example, CSGM-Langevin's performance is significantly reduced when the measurement gradient step size increases to $7.5$ or $10$, while DiffStateGrad's performance decline is less. This suggests that DiffStateGrad can be a practical tool to reduce the risk of drastic failure in real-world applications. Finally, given the small sample size and the preliminary nature of this analysis, we leave extensive analysis for future work.

### E.1 NYU FASTMRI DATASET

The data used in the above experiment was obtained from the NYU fastMRI Initiative database (`fastmri.med.nyu.edu`) (Zbontar et al., 2018; Knoll et al., 2020). We note that NYU fastMRI investigators provided data but did not participate in the analysis or writing of this report. See their website for more information on the primary goal of fastMRI to advance machine learning research for the reconstruction of medical images.

