# OpenReview forum: "Diffusion State-Guided Projected Gradient for Inverse Problems"
_ICLR.cc/2025/Conference — ICLR 2025 Poster_

### Official Review · Reviewer_3FsW · 2024-10-21

**Soundness:** 2
**Presentation:** 2
**Contribution:** 2
**Rating:** 3
**Confidence:** 4

**Summary:**

The authors propose DiffStateGrad, a method for projecting the gradients to a low-rank subspace through SVD, where the gradients are those arising from the diffusion models for inverse problem-solving (DIS) framework. The authors propose an adaptive thresholding value, where it is automatically calculated for every timestep $t$. It is shown that DiffStateGrad improves the performance compared to when it is not used. DiffStateGrad is shown to be especially useful for latent diffusion-based methods, which is understandable as gradients are noisier due to the existence of the decoder.

**Strengths:**

1. The method is straightforward to understand and easy to implement.

2. The proposed framework can be used regardless of the base framework. In the three frameworks that the authors tested DiffStateGrad, it always shows improved performance.

**Weaknesses:**

1. I believe DiffStateGrad should mainly be compared to approaches like MPGD [1] or DDS [2], which are highly related to this work. The authors do mention [1] as a related work, but says that DiffStateGrad is different to [1], as it considers a projection on $\mathcal{M}_t$ and not $\mathcal{M}$. However, care should be taken because applying gradient guidance to the posterior mean or $x_t$ are equivalent up to a constant. Hence, projecting the gradient guidance on MPGD has about the same effect. The same goes for [2], which does not explicitly project the gradient but uses a Krylov subspace method. I believe DiffStateGrad should be directly compared to MPGD.

2. The fact that guidance on the posterior mean and the guidance on $x_t$ only differs by a scale factor questions if the theory presented in the paper is actually useful.

3. Most of the experiments are conduced with latent diffusion based methods. More experiments should be conducted on other widely used pixel domain methods such as DPS and $\Pi$GDM.




**References**

[1] He, Yutong, et al. "Manifold preserving guided diffusion." ICLR 2024

[2] Chung, Hyungjin, et al., "Decomposed diffusion sampler for accelerating large-scale inverse problems", ICLR 2024

**Questions:**

Please see weaknesses.

---

> ### Author Response · Authors · 2024-11-21
> **Authors' Initial Response (1)**
>
> We thank the reviewer for their thoughtful comments. First, we would like to refer the reviewer to our general summary of response post. The post summarizes the key points and additional experiments that we have now included in the paper. Below, we provide our response to each of the reviewer's comments.
>
> **1. [Comparison of DiffStateGrad projection to other projection methods]**
>
> We thank the reviewer for raising this point. We have included additional explanations and experimental results on explaining how our method differs from MPGD [1] and MCG [2].
>
> - MPGD [1] employs a manifold-preserving approach to enhance the efficiency of diffusion generation and solve inverse problems. While this method aligns with a similar motivation as our proposed framework, it is specifically designed for cases where the measurement gradient is applied to $x_{0|t}$ or $z_{0|t}$. In this scenario, MPGD suggests projecting the gradient onto the tangent space of the clean data manifold. Its most effective implementation, MPGD-AE, utilizes an autoencoder (AE) for this manifold projection. However, this approach is limited by the need for an autoencoder, and its performance heavily depends on the expressive power of the autoencoder used. In contrast, DiffStateGrad is more versatile, as it can be applied to methods that use guidance on $z_{0|t}$/$x_{0|t}$ (e.g., ReSample [7] and DAPS [6]) as well as methods that apply the guidance to $x_{t}$/$z_{t}$ (e.g., PSLD [8] and DPS [1]). Notably, MPGD does not support the latter scenario. For the former scenario, we agree with a reviewer that both approaches of MPGD and DiffStateGrad can be compared and the comparison can highlight the effectiveness of the choice of projection step. Our approach constructs the projection using SVD to capture the data structure and does not need a trained AE. We have clarified this distinction in the revised manuscript and included additional explanations in the Related Works section. We conclude that our results from new Tables 3 and 4 confirm that among pixel-based methods using the guidance to update $x_{0|t}$, DiffStateGrad-DAPS outperforms DAPS and MPGD-AE. Moreover, while MPGD achieves LPIPS metrics that are comparable to or slightly better than DAPS for FFHQ, its PSNR performance is significantly lacking. To further highlight MPGD-AE's dependency on the expressive power of the autoencoder (AE) used, we refer the reviewer to our new results on MPGD-AE for inpainting tasks on ImageNet, where MPGD-AE performs very poorly compared to both DAPS and our DiffStateGrad-DAPS.
>
> - We agree with the reviewer that projecting onto the clean data manifold for $x_{0|t}$ is equivalent to projecting onto $M_t$ of $x_t$ up to a constant. We clarify that the key aspect of our paper is that the projection should be defined based on the variable to which the gradient guidance is being applied. For example, if the guidance is applied to $x_{0|t}$, the projection should be defined based on $x_{0|t}$; this is similar to the intuition behind MPGD, and is applicable to methods such as ReSample and DAPS. Moreover, if the guidance is applied to $x_t$, the projection should be defined based on $x_t$. This latter scenario is applicable to approaches like DPS and PSLD. Importantly, our DiffStateGrad framework offers performance improvements in both cases.
>
> - Lastly, we would like to refer the reviewer to our general response G2, where we include additional explanations on how our method differs from MCG, which also uses a manifold projection concept.
>
> **2. [Theory]**
>
> We thank the reviewer for raising this important point, as elaborating on it helps to improve the clarity and quality of the paper. The theoretical analysis in this work supports the argument that the measurement gradient should be projected onto the tangent space of the manifold at time $t$ (if the gradient update is applied to $z_t$ or $x_t$) or at time $0$, corresponding to the clean data manifold (if the gradient update is applied to $z_{0|t}$ or $x_{0|t}$).
>
> This intuition aligns with the reasoning discussed in MPGD [1], which advocates for projecting the gradient onto the tangent space of the clean data manifold. While our analysis primarily formulates the problem for scenarios where the measurement gradient is applied to update $z_t$, we have now clarified in the paper that, in cases where the gradient update is applied to $z_{0|t}$, the projection should be onto the tangent space of $z_{0|t}$. This latter scenario is indeed used for DiffStateGrad-ReSample and DiffStateGrad-DAPS, and it is analogous to the approach taken in MPGD. We hope that this clarification has addressed the reviewer's concern.

---

> > ### Author Response · Authors · 2024-11-21
> > **Authors' Initial Response (2)**
> >
> > **3. [Pixel-based experiments]**
> >
> > In response to the reviewer's valuable suggestions, we have conducted additional experiments with "pixel-based" diffusion models and included the results in the revised manuscript. These experiments demonstrate that DiffStateGrad is not only applicable but also scalable to both latent- and pixel-based diffusion models. The results confirm that DiffStateGrad consistently enhances performance and outperforms prior work in the literature when applied to both pixel-based and latent-based models that use a guidance term to solve inverse problems.
> >
> > We have combined the previous Tables 2, 3, and 4 into a new Table 3, which now includes results for all linear and nonlinear inverse problems. In addition, we conducted new experiments for DPS, DiffStateGrad-DPS, and MPGD-AE, and reported results for DDNM, DDRM, and MCG on linear inverse problems using the FFHQ dataset. Furthermore, we included new pixel-based quantitative results in new Table 4a, showcasing comparisons among DAPS, DiffStateGrad-DAPS, DPS, DiffStateGrad-DPS, and MPGD-AE for box inpainting and random inpainting on ImageNet. Lastly, Table 4b presents additional results for DAPS, DiffStateGrad-DAPS, and MCG on Gaussian deblurring, motion deblurring, and super-resolution inverse problems conducted on ImageNet.
> >
> >
> > The results demonstrate that DiffStateGrad improves the performance and is superior to other projection-based diffusion models such as MCG and MPGD-AE. We include a summary of these results below (please also see our G1 general response to all reviewers).
> >
> > **New Results in Table 3: Performance comparison of linear tasks on FFHQ 256 x 256.**
> >
> > | Method                        | LPIPS ↓ (Box-In) | PSNR ↑ (Box-In) | LPIPS ↓ (Random-In) | PSNR ↑ (Random-In) | LPIPS ↓ (Gaussian-Db) | PSNR ↑ (Gaussian-Db) | LPIPS ↓ (Motion-Db) | PSNR ↑ (Motion-Db) | LPIPS ↓ (SR) | PSNR ↑ (SR) |
> > |-------------------------------|---------------|--------------|------------------|-----------------|--------------------|-------------------|------------------|-----------------|--------------|-------------|
> > | **Pixel-based**               |               |              |                  |                 |                    |                   |                  |                 |              |             |
> > | DAPS                          | 0.136         | 24.57        | 0.130            | 30.79           | 0.216              | 27.92             | 0.154            | 30.13           | 0.197        | 28.64       |
> > | DiffStateGrad-DAPS (ours)     | **0.113**     | **24.78**    | **0.099**        | **32.04**       | 0.180          | **29.02**         | 0.119        | **31.74**       | 0.181    | **29.35**   |
> > | DPS                           | 0.127         | 23.91        | 0.130            | 28.67           | 0.145              | 25.48             | 0.132            | 26.75           | 0.191        | 24.38       |
> > | DiffStateGrad-DPS (ours)      | 0.114     | 24.10    | 0.107        | 30.15      | **0.128**          | 26.29         | **0.118**        | 27.61      | 0.186    | 24.65   |
> > | DDNM                          | 0.235         | 24.47        | 0.121            | 29.91           | 0.216              | 28.20             | -                | -               | 0.197        | 28.03       |
> > | DDRM                          | 0.159         | 22.37        | 0.218            | 25.75           | 0.236              | 23.36             | -                | -               | 0.210        | 27.65       |
> > | MCG                           | 0.309         | 19.97        | 0.286            | 21.57           | 0.340              | 6.72              | 0.702            | 6.72            | 0.520        | 20.05       |
> > | MPGD-AE                       | 0.138         | 21.59        | 0.172            | 25.22           | 0.150              | 24.42             | 0.120            | 25.72           | **0.168**    | 24.01   |

---

> > > ### Author Response · Authors · 2024-11-21
> > > **Authors' Initial Response (3)**
> > >
> > > **New Results in Table 4: Performance comparison of linear tasks on ImageNet 256 x 256.**
> > >
> > > **Inpainting tasks.**
> > >
> > > | Method                        | LPIPS ↓ (Box-In)      | SSIM ↑ (Box-In)       | PSNR ↑ (Box-In)       | LPIPS ↓ (Random-In)   | SSIM ↑ (Random-In)   | PSNR ↑ (Random-In)   |
> > > |-------------------------------|--------------------|--------------------|--------------------|--------------------|-------------------|-------------------|
> > > | DAPS                          | 0.217 (0.043)      | 0.762 (0.041)      | 20.90 (3.69)       | 0.158 (0.039)      | 0.794 (0.067)     | 28.34 (3.65)      |
> > > | DiffStateGrad-DAPS (ours)     | **0.191** (0.044)  | **0.801** (0.056)  | **21.07** (3.77)   | **0.107** (0.037)  | **0.856** (0.067) | **29.78** (4.17)  |
> > > | DPS                           | 0.257 (0.086)      | 0.718 (0.097)      | 19.85 (3.54)       | 0.256 (0.133)      | 0.728 (0.143)     | 26.25 (4.15)      |
> > > | DiffStateGrad-DPS (ours)      | 0.243 (0.093)  | 0.731 (0.100)  | 19.87 (3.61)   | 0.233 (0.138)  | 0.754 (0.150) | 27.28 (4.88)  |
> > > | MPGD-AE                       | 0.295 (0.057)      | 0.621 (0.053)      | 16.12 (2.26)       | 0.554 (0.148)      | 0.388 (0.112)     | 17.91 (3.25)      |
> > >
> > > **Deblurring and super-resolution tasks.**
> > >
> > > | Method                        | LPIPS ↓ (Gaussian-Db) | PSNR ↑ (Gaussian-Db)   | LPIPS ↓ (Motion-Db)   | PSNR ↑ (Motion-Db)   | LPIPS ↓ (SR)      | PSNR ↑ (SR)      |
> > > |-------------------------------|--------------------|---------------------|--------------------|-------------------|-------------------|------------------|
> > > | DAPS                          | 0.266      | 25.27        | 0.166       | 28.85     | 0.259   | 25.67     |
> > > | DiffStateGrad-DAPS (ours)     | **0.243**  | **25.87**    | **0.143** | **29.71**  | **0.229**  | **26.40**  |
> > > | MCG                           | 0.550         | 16.32           | 0.758           | 5.89          | 0.637         | 13.39        |
> > >
> > >
> > > **New Results in Table 6: SSIM comparison of linear tasks on FFHQ 256 x 256.**
> > >
> > > | Method                        | SSIM ↑ (Box-In)       | SSIM ↑ (Random-In)    | SSIM ↑ (Gaussian-Db)  | SSIM ↑ (Motion-Db)    | SSIM ↑ (SR)       |
> > > |-------------------------------|--------------------|--------------------|--------------------|--------------------|-------------------|
> > > | DAPS                          | 0.806 (0.028)      | 0.829 (0.022)      | 0.786 (0.051)      | 0.837 (0.040)      | 0.797 (0.044)     |
> > > | DiffStateGrad-DAPS (ours)     | **0.849** (0.029)  | **0.887** (0.023)  | **0.803** (0.044)  | **0.853** (0.028)  | **0.801** (0.039) |
> > > | DPS                           | 0.810 (0.039)      | 0.815 (0.045)      | 0.709 (0.062)      | 0.754 (0.056)      | 0.675 (0.071)     |
> > > | DiffStateGrad-DPS (ours)      | 0.831 (0.039)  | 0.852 (0.046)  | 0.739 (0.062)  |  0.782 (0.056)  | 0.683 (0.073) |
> > > | MPGD-AE                       | 0.753 (0.029)      | 0.731 (0.050)      | 0.664 (0.071)      | 0.723 (0.061)      | 0.670 (0.070)     |
> > >
> > > [1] He, Y., Murata, N., Lai, C. H., Takida, Y., Uesaka, T., Kim, D., ... & Ermon, S. Manifold Preserving Guided Diffusion. In The Twelfth International Conference on Learning Representations.
> > >
> > > [2] Chung, H., Sim, B., Ryu, D., & Ye, J. C. (2022). Improving diffusion models for inverse problems using manifold constraints.
> > >
> > > [3] Chung, H.d , Kim, J., Mccann, M. T., Klasky, M. L., & Ye, J. C. Diffusion Posterior Sampling for General Noisy Inverse Problems. In The Eleventh International Conference on Learning Representations.
> > >
> > > [4] Song, B., Kwon, S. M., Zhang, Z., Hu, X., Qu, Q., & Shen, L. Solving Inverse Problems with Latent Diffusion Models via Hard Data Consistency. In The Twelfth International Conference on Learning Representations.
> > >
> > > [5] Zhang, B., Chu, W., Berner, J., Meng, C., Anandkumar, A., & Song, Y. (2024). Improving diffusion inverse problem solving with decoupled noise annealing. arXiv preprint arXiv:2407.01521.

---

> > > > ### Author Response · Authors · 2024-11-21
> > > > **Authors' Initial Response (4)**
> > > >
> > > > We hope our response to the reviewer's comments as well as our response to other reviewers along with the general response addresses the reviewer's concerns, and we remain open to further feedback. We look forward to the reviewer's opinion on our newly improved submission.

---

> > > > > ### Author Response · Authors · 2024-11-25
> > > > > **Authors are looking forward to hearing from reviewer 3FsW**
> > > > >
> > > > > The authors are looking forward to hearing from reviewer 3FsW. We believe that the reviewer's feedback has improved our manuscript. We would like to know if our modifications, responses, clarifications, and new experimental results address the reviewer's concerns. We welcome further feedback.

---

> > > > > > ### Comment · Reviewer_3FsW · 2024-11-27
> > > > > >
> > > > > > I would like to thank the authors for the complete and thorough response. Some of my concerns have been resolved. Although I appreciate the thorough experiments and notice the positive comments from the other reviewers, I am a bit skeptical. Below are my concerns:
> > > > > >
> > > > > > 1. When comparing against MPGD, it should be compared in the same manner as DiffStateGrad. Specifically, MPGD can also be thought of as a plug-in method that would apply to all gradient-based guidance methods. For instance, a fair comparison of DiffStateGrad-DAPS would be MPGD-DAPS, **not** MPGD-AE.
> > > > > >
> > > > > > 2. Could you also provide comments or experiments against DDS, which was stated in my earlier comments?
> > > > > >
> > > > > > 3. If the projection on $\mathcal{M}$ and $\mathcal{M_t}$ only differs by a constant, I agree that it should be defined w.r.t. to the variable of interest, but this does not mean that it would have a different effect.

---

> > > > > > > ### Author Response · Authors · 2024-11-28
> > > > > > > **Authors' Response to 1.**
> > > > > > >
> > > > > > > We sincerely thank the reviewer for their thoughtful question. Below, we address the limitations of MPGD [1] compared to DiffStateGrad and provide clarification on how our reported results should be interpreted.
> > > > > > >
> > > > > > > **1. Limitations of MPGD**
> > > > > > >
> > > > > > > - Dependence on autoencoder (AE): MPGD requires a pre-trained autoencoder that is trained on a clean data distribution. Therefore, MPGD is not applicable in scenarios where such a trained AE is unavailable.
> > > > > > > - Sensitivity to hyperparameters: The projection in MPGD involves scaling hyperparameters related to the gradient step size, which require careful tuning to achieve good performance. While we tuned this parameter when reporting results, the sensitivity of MPGD to the scaling step size adds to its limitations. To illustrate this limitation, we conducted a small-scale experiment for super-resolution on 10 random examples from the FFHQ dataset, varying the scaling hyperparameter. The results are summarized below, demonstrating that MPGD is not robust to changes in the gradient step size.
> > > > > > >
> > > > > > > | Method                        | PSNR ↑ | SSIM ↑ |  LPIPS ↓ |
> > > > > > > |-------------------------------|--------------|------------------|------------------|
> > > > > > > MPGD (scale = 0.25) | 21.5836 (1.6875) | 0.5847 (0.0680) | 0.2432 (0.0499) |
> > > > > > > MPGD (scale = 0.5 optimal)  | 24.4700 (1.7766) | 0.6854 (0.0499) | 0.1535 (0.0356) |
> > > > > > > MPGD (scale = 1) | 24.7273 (1.7350) | 0.6523 (0.0420) | 0.1956 (0.0619) |
> > > > > > > MPGD (scale = 2) | 22.9427 (1.3345) | 0.4733 (0.0443) | 0.3735 (0.0738) |
> > > > > > >
> > > > > > > In contrast, DiffStateGrad does not require a training procedure. Additionally, as we previously demonstrated in Figure 5, DiffStateGrad improves robustness to gradient step size, and Figure 9 further highlights its robustness to a wide range of variance retention thresholds. This makes DiffStateGrad a more practical and reliable approach.
> > > > > > >
> > > > > > >
> > > > > > > **2. Comparison of DiffStateGrad to MPGD**
> > > > > > >
> > > > > > > - Extent of analysis in MPGD: To our knowledge, the literature lacks extensive quantitative analyses of MPGD. The primary focus of [1] is on acceleration (using fewer DDIM steps) and a wide range of conditioning tasks beyond vision restoration inverse problems. Moreover, we are unaware of any recent state-of-the-art work that compares to MPGD extensively. Additionally, [1] does not evaluate MPGD as a plug-in method. To ensure a fair comparison, we used the official MPGD implementation (based on DPS [2]) and tuned its scaling hyperparameter for optimal performance. Consequently, our comparisons are conducted between DPS, MPGD, and DiffStateGrad-DPS. Our results show that DiffStateGrad-DPS is superior. While MPGD could theoretically be applied to DAPS, we do not compare against MPGD-DAPS due to the absence of an officially published implementation. However, we speculate that DiffStateGrad-DAPS would similarly outperform a hypothetical MPGD-DAPS.
> > > > > > >
> > > > > > > - Clarification on naming: The term "MPGD-AE" specifically refers to the version of MPGD that uses an autoencoder for projection. To avoid confusion, we have modified the naming in our manuscript to simply "MPGD."
> > > > > > >
> > > > > > > - Reproducibility and additional experiments: To support the credibility of our reported results, we conducted a supplementary experiment on box inpainting for the FFHQ dataset using reduced DDIM steps (250) instead of the original 1000. This demonstrates our ability to reproduce the statements in [1], where MPGD outperforms DPS when fewer DDIM steps are used. However, our results suggest that this advantage diminishes when a larger number of DDIM steps (1000) are used, a scenario not discussed in [1].
> > > > > > >
> > > > > > > DDIM steps = 250
> > > > > > > | Method                        | PSNR ↑ | SSIM ↑ |  LPIPS ↓ |
> > > > > > > |-------------------------------|--------------|------------------|------------------|
> > > > > > > MPGD | 20.78 | 0.746 |  0.1447 |
> > > > > > > DPS | 17.66 | 0.710 | 0.2670 |
> > > > > > > DiffStateGrad-DPS (Ours) | 17.73 | 0.751 | 0.2565 |
> > > > > > >
> > > > > > > DDIM steps = 1000
> > > > > > > | Method                        | PSNR ↑ | SSIM ↑ |  LPIPS ↓ |
> > > > > > > |-------------------------------|--------------|------------------|------------------|
> > > > > > > MPGD | 21.59 | 0.753 | 0.138 |
> > > > > > > DPS | 23.91 | 0.810 | 0.127 |
> > > > > > > DiffStateGrad-DPS (Ours) | 24.10 | 0.831 | 0.114 |
> > > > > > >
> > > > > > > These results confirm the findings of [1] that MPGD outperforms DPS when a low number of DDIM steps is used. However, our results demonstrate that with 1000 DDIM steps (the standard for reporting DPS and MPGD results in our work), MPGD loses its main advantage compared to DPS. This further highlights that DiffStateGrad-DPS consistently outperforms both DPS and MPGD.
> > > > > > >
> > > > > > > We hope this detailed explanation addresses the reviewer's concerns and provides clarity on the scope and methodology of our comparisons. Thank you again for your valuable feedback.

---

> > > > > > > > ### Author Response · Authors · 2024-11-28
> > > > > > > > **Authors' Response to 2.**
> > > > > > > >
> > > > > > > > We thank the reviewer for pointing us to DDS. The Decomposed Diffusion Sampler (DDS) framework proposes replacing the MCG manifold constraint, which is computationally expensive, with Conjugate Gradient (CG) to produce a more efficient measurement gradient aimed at inverting the forward model $y = Ax$.
> > > > > > > >
> > > > > > > > In the context of Table 2, DDS uses the denoised sample $x_{0|t}$, updates it with Conjugate Gradient, and then resamples to step $t$ of the diffusion. However, unlike ours, DDS is not a plug-in method, and not applicable to approaches such as PSLD/DPS where the gradient is used to update the state $t$. Instead, it employs a decomposed sampling and measurement consistency strategy. Additionally, DDS relies on Conjugate Gradient, which requires the adjoint of the forward operator $A$. Consequently, DDS is only applicable to linear inverse problems where $A^{*}$ is easily computable. This limitation is evident in their experiments, which focus solely on MRI reconstruction.
> > > > > > > >
> > > > > > > > In summary, DDS is not suitable for nonlinear inverse problems, and its publication provides an analysis restricted to a single inverse problem task (MRI). In contrast, our manuscript evaluates DiffStateGrad across a wide range of both linear and nonlinear image restoration tasks, highlighting its broader applicability. Given the reviewer's suggestion on this comparison, we will cite DDS and provide above in Related Works.

---

> > > > > > > > > ### Author Response · Authors · 2024-11-28
> > > > > > > > > **Authors' Response to 3.**
> > > > > > > > >
> > > > > > > > > We thank the reviewer for their thoughtful engagement in this discussion. We agree with the reviewer that, in theory and under ideal settings, the methods may have a similar impact. However, in practice, differences can arise depending on how well the projection operator is defined and how the projection is applied.
> > > > > > > > >
> > > > > > > > > We would like to clarify that performing projections with respect to the variable of interest is not yet a fully established step in the literature. As discussed in our revised manuscript, MCG [3] projects onto $\mathcal{M}_0$ while the gradient updates $x_t$. We agree with the reviewer that both MPGD and DiffStateGrad align in philosophy by recommending projection onto $\mathcal{M}0$ when $x{0|t}$ is being updated. While these methods share this underlying sentiment, we highlight the following differences, which may contribute to the robustness and superiority of DiffStateGrad:
> > > > > > > > >
> > > > > > > > > 1. Projection Mechanism:
> > > > > > > > >
> > > > > > > > > - MPGD (particularly MPGD-AE) uses an autoencoder to project $x_{0|t}$ onto a manifold and then computes the measurement gradient using the output of the autoencoder. The projected $x_{0|t}$ is subsequently updated.
> > > > > > > > > - In contrast, DiffStateGrad computes the measurement gradient using $x_{0|t}$ directly and then projects the gradient onto a subspace before performing likelihood-related gradient descent.
> > > > > > > > >
> > > > > > > > > We speculate that DiffStateGrad's robustness and improved performance stem from two key factors: the choice of projection subspace and the projection of the gradient rather than the diffusion state. While these differences may seem subtle, they appear to have a noticeable impact on performance.
> > > > > > > > >
> > > > > > > > > 2. Performance Superiority:
> > > > > > > > >
> > > > > > > > > Our earlier results comparing MPGD, DPS, and DiffStateGrad-DPS clearly demonstrate that DiffStateGrad achieves better performance than MPGD in practice.
> > > > > > > > >
> > > > > > > > > [1] He, Y., Murata, N., Lai, C. H., Takida, Y., Uesaka, T., Kim, D., ... & Ermon, S. Manifold Preserving Guided Diffusion. In The Twelfth International Conference on Learning Representations.
> > > > > > > > >
> > > > > > > > > [2] Chung, H., Kim, J., Mccann, M. T., Klasky, M. L., & Ye, J. C. Diffusion Posterior Sampling for General Noisy Inverse Problems. In The Eleventh International Conference on Learning Representations.
> > > > > > > > >
> > > > > > > > > [3] Chung, H., Sim, B., Ryu, D., & Ye, J. C. (2022). Improving diffusion models for inverse problems using manifold constraints. Advances in Neural Information Processing Systems, 35, 25683-25696.

---

> > > > > > > > > > ### Author Response · Authors · 2024-11-28
> > > > > > > > > >
> > > > > > > > > > We look forward to hearing the reviewer's thoughts on our clarifications and new analyses.
> > > > > > > > > >
> > > > > > > > > > We hope that the explanations provided above address the reviewer’s concerns. If so, we kindly request the reviewer to reflect this in their score. If there are any remaining concerns, we would greatly appreciate further feedback on how we might address them.

---

> > > > > > > > > > > ### Author Response · Authors · 2024-12-01
> > > > > > > > > > >
> > > > > > > > > > > As the discussion period is ending, we haven't heard from the reviewer on our latest response. We kindly ask the reviewer to read our above response, including new experiments and clarification, and let us know if a concern is left unaddressed.
> > > > > > > > > > >
> > > > > > > > > > > We look forward to hearing from the reviewer.

---

### Official Review · Reviewer_kB1E · 2024-11-04

**Soundness:** 3
**Presentation:** 3
**Contribution:** 3
**Rating:** 8
**Confidence:** 4

**Summary:**

This paper tackles the problem of solving from a posterior distribution defined with a diffusion-based prior. The authors assume that a diffusion model has been pre-trained and then use it to solve various inverse problems while not requiring anymore training. The method proposed can be thought of as enhancement for existing solvers; it can applied on top of any existing diffusion-based posterior sampling algorithm. It basically consists in adding a projection step, where the projection operator is computed based on the current diffusion state.

**Strengths:**

- The paper is well written and the methodology is clearly presented. The fact that the proposed algorithm can be plugged on top of existing methods is a significant feat. The theoretical justification for the method is loose but the explanations provided are intuitive.
- The experiments are convincing are well thought and rather extensive.

**Weaknesses:**

- While the additional computational cost is marginal for latent diffusion models, isn't it prohibitive for pixel space diffusion models? Having to compute the SVD at each iteration is certainly a significant drawback.
- The methodology is based on having at some point a sample on the manifold of "artifact free images" (this is loosely defined). It is unclear how this arises in practice.
- There are some inconsistencies in the experiments which I believe are explained nowhere in the paper. First, the authors compare to DAPS in pixel space but not in latent space, although in the original paper DAPS is applied in latent space too. Second, in the only example involving imagenet, the authors compare to PSLD only and not ReSample nor PSLD. Also, no experiment on pixel space imagenet is provided. Is there any reason for this?

**Questions:**

- While it is claimed that you test the robustness of the method to measurement noise, I am failing where this is done in the paper.
- How would the method apply to more general methods, e.g. to methods that do not use the approximate guidance term, for example [1]. Would you then simply project the updated $x_t$ and reconstruct it, then move on to the next step?

[1] Lugmayr, Andreas, Martin Danelljan, Andres Romero, Fisher Yu, Radu Timofte, and Luc Van Gool. "Repaint: Inpainting using denoising diffusion probabilistic models." In Proceedings of the IEEE/CVF conference on computer vision and pattern recognition, pp. 11461-11471. 2022.

---

> ### Author Response · Authors · 2024-11-21
> **Authors' Initial Response (1)**
>
> We thank the reviewer for their thoughtful comments and appreciation of our extensive experimental results and intuitions on the proposed method. We kindly refer the reviewer to our general comment addressed to all reviewers, where we summarize the new experiments we conducted and highlight the key points of our work. Below, we provide a detailed response to the reviewer's specific concerns, addressing issues related to more experiments on pixel-based diffusion models, the intuition on the artifact-free images, latent DAPS, robustness to measurement noise, and comparison to gradient-free methods such as RePaint.
>
> **1. [Pixel-based Diffusion Models]**
>
> We thank the reviewer for raising this concern.
>
> - We now include extensive additional pixel-based diffusion models in Tables 3 and 4 (please see our G1 general response in the comment to all reviewers). The results confirm that DiffStateGrad improves performance. This supports the scalability and applicability of DiffStateGrad to pixel-based diffusion models.
>
> - Moreover, we note that our implementation includes a projection period parameter, which allows SVD computations and projections to be performed less frequently. For instance, while our experiments with DAPS apply projections at every iteration, the ReSample experiments use projections every five iterations. This information has been included in the Appendix for reference. We have conducted an additional set of experiments, where DiffStateGrad is applied to DAPS every 5 or 10 iterations. The results which we report below suggest that DiffStateGrad can be applied at less frequent manner for efficiency purposes and still improve performance upon the DAPS without projection. See our response in G3 for more detailed information. These results can be found in the new Figure 10 in the Appendix. Overall, the results, summarized below, demonstrate the flexibility and efficiency of our approach.
>
>
> | Method                        | LPIPS ↓ | SSIM  ↑ |  PSNR ↑ |
> |-------------------------------|---------------|---------------|---------------|
> | DAPS                          |  0.130   |  0.829  |  30.79  |
> | DiffStateGrad-DAPS (ours) P=1    | **0.099**  |  **0.887**  |   **32.04** |
> | DiffStateGrad-DAPS (ours) P=5   | 0.116   | 0.863   |    31.41 |
> | DiffStateGrad-DAPS (ours) P=10   |  0.118  |  0.860   |  31.36 |
>
> where P denotes the projection period.
>
> - Lastly, we kindly refer the reviewer to our efficiency analysis presented in Figure 4, where we demonstrate that the additional runtime complexity introduced by DiffStateGrad is minimal relative to the overall runtime of the diffusion solver. In this analysis, DAPS, a pixel-based method where the projection happens as every iteration (P=1). We reason that for P>1, this runtime should even be more negligible.
>
> Overall, we deeply appreciate the reviewer's insightful concern and acknowledge that future work on DiffStateGrad could explore more efficient implementations of SVD. Additionally, while we currently use SVD to identify a low-rank structure of the diffusion state ($x_{t}$ for DPS, $x_{0|t}$ for DAPS), future research could investigate alternative approaches for identifying subspaces that effectively represent the diffusion state and the local manifold structure for projection purposes.
>
>
> **2. [Artifact-free images]**
>
> The literature on using diffusion models to solve inverse problems operates under the assumption that there exists a diffusion model unconditionally trained on a dataset (e.g., clean images). The term "artifact-free" refers to the generation process of such an unconditional diffusion model, trained on clean data samples, which provides an artifact-free trajectory from $\mathcal{M}_T$ to $\mathcal{M}_0$. For further intuition on projection onto the tangent space of the clean data manifold, we refer the reviewer to MPGD [4].
>
> In our work, the reference to artifact-free images pertains specifically to this learned path, which captures general "clean and artifact-free" images, rather than blurred images or images with missing pixels. The intuition behind our framework is to define a projection step that minimizes movements caused by the measurement gradient that are orthogonal to the manifold locally, either at $x_t$ (if the measurement guidance is applied to $x_t$) or at $x_{0|t}$ (if the measurement guidance is applied to $x_{0|t}$).
>
> Focusing on the former case (without loss of generality), given a single sample $x_t$ from the manifold $\mathcal{M}_t$, we use $x_t$ to approximate and capture the local structures of $\mathcal{M}_t$. These points have now been included in the revised manuscript, and we hope this explanation clarifies the reviewer’s comment.

---

> > ### Author Response · Authors · 2024-11-21
> > **Authors' Initial Response (2)**
> >
> > **3. [Latent-DAPS]**
> >
> > We thank the reviewer for their thoughtful comment.
> >
> > - Our choice to use DAPS instead of Latent-DAPS was due to the unavailability of Latent-DAPS code in the authors' public GitHub repository, which only provides an implementation of DAPS. Furthermore, we note that ReSample is a special case of Latent-DAPS, as discussed in [5]. Hence, we speculate that our method should improve Latent-DAPS, given the superior performance of DiffStateGrad-ReSample against ReSample.
> >
> > - Regarding the initial lack of additional results, this was purely due to time constraints. In response to the reviewer's feedback, we have now conducted new experiments and included extensive pixel-based comparisons. The new Tables now contain quantitative results from DPS, DiffStateGrad-DPS, DAPS, DiffStateGrad-DAPS, MPGD-AE, DDNM, DDRM, and MCG. We refer the reviewer to G1 in our general response and also the revision of the manuscript for details. The results consistently demonstrate that DiffStateGrad improves performance for both pixel-based and latent-based methods across a wide range of linear and nonlinear inverse problems on two datasets (FFHQ and ImageNet).
> >
> > We hope this response addresses the reviewer's concerns, and we remain open to further feedback.
> >
> > **4. [Robustness to Measurement Noise]**
> >
> > Figure 6 presents the results on the impact of measurement noise on performance. In Figure 6a, we focus on a pixel-based diffusion scenario for the box inpainting task, where DiffStateGrad-DAPS (blue solid line) demonstrates significantly greater robustness compared to DAPS (red dashed line) as the measurement noise std (x-axis) increases, measured in terms of SSIM (y-axis).
> >
> > Similarly, Figure 6b illustrates the performance of a latent-based diffusion model on the Gaussian deblurring task. As the measurement noise std (x-axis) increases, the performance of PSLD declines at a much faster rate compared to DiffStateGrad-PSLD, highlighting the improved robustness introduced by DiffStateGrad.
> >
> > These results are summarized and discussed in Section 4.2, "Robustness." To enhance clarity, we have updated the x-axis label from "Measurement Noise Level" to "Measurement Noise Std." Moreover, suggested by another reviewer, we have extended the x-axis to the noiseless setting (measurement noise std = 0).
> >
> > **5. [Comparison to Repaint]**
> >
> > We thank the reviewer for pointing us to the RePaint method. We now cite the paper in related works, and explain the applicability of RePaint as follows in the paper:
> >
> > - "Finally, we note that while this work focuses on gradient-based guidance, prior work such as RePaint introduces a gradient-free masking strategy to solve inverse problems. Although RePaint is appealing, it is limited to inpainting tasks and scenarios where measurement noise is negligible."
> >
> > Hence, RePaint is not applicable to general inverse problems such as deblurring, super-resolution, phase retrieval, and others. Furthermore, it has been reported in [1] that Manifold Constrained Gradient (MCG) outperforms RePaint on inpainting tasks. Our new experimental results demonstrate that the DiffStateGrad projection applied to DPS outperforms MCG, which is a special case of DPS [3]. Based on this, we speculate that DiffStateGrad is also superior to RePaint.

---

> > > ### Author Response · Authors · 2024-11-25
> > > **Authors are looking forward to hearing from reviewer kB1E**
> > >
> > > We would like to thank again the reviewer for their comments. We believe that their feedback, which has now been addressed in the revised manuscript, has increased the quality of the paper. We appreciate the reviewer's response to our rebuttal and if there is an additional point that needs clarification. We remain open to feedback.

---

> > > > ### Comment · Reviewer_kB1E · 2024-11-26
> > > >
> > > > I would like to thank the authors for the significant work they have done during the rebuttal period. I believe that my concerns have been addressed. I raise my score

---

> > > > > ### Author Response · Authors · 2024-11-28
> > > > >
> > > > > We sincerely thank the reviewer for raising their score and providing valuable feedback. We greatly appreciate their thoughtful review, which we believe has significantly contributed to improving the quality of our manuscript.

---

> > ### Author Response · Authors · 2024-11-21
> > **Authors' Initial Response (3)**
> >
> > We hope our response addresses the reviewer's comments; if not, we welcome additional discussion.
> >
> > ---
> > References
> >
> > [1] Chung, H., Sim, B., Ryu, D., & Ye, J. C. (2022). Improving diffusion models for inverse problems using manifold constraints. Advances in Neural Information Processing Systems, 35, 25683-25696.
> >
> > [2] Lugmayr, A., Danelljan, M., Romero, A., Yu, F., Timofte, R., & Van Gool, L. (2022). Repaint: Inpainting using denoising diffusion probabilistic models. In Proceedings of the IEEE/CVF conference on computer vision and pattern recognition (pp. 11461-11471).
> >
> > [3] Chung, H, Kim, J., Mccann, M. T., Klasky, M. L., & Ye, J. C. Diffusion Posterior Sampling for General Noisy Inverse Problems. In The Eleventh International Conference on Learning Representations.
> >
> > [4] He, Y., Murata, N., Lai, C. H., Takida, Y., Uesaka, T., Kim, D., ... & Ermon, S. Manifold Preserving Guided Diffusion. In The Twelfth International Conference on Learning Representations.
> >
> > [5] Zhang, B., Chu, W., Berner, J., Meng, C., Anandkumar, A., & Song, Y. (2024). Improving diffusion inverse problem solving with decoupled noise annealing. arXiv preprint arXiv:2407.01521.

---

### Official Review · Reviewer_D85G · 2024-11-04

**Soundness:** 4
**Presentation:** 4
**Contribution:** 3
**Rating:** 8
**Confidence:** 4

**Summary:**

Authors propose Diffusion State-Guided Projected Gradient (DiffStateGrad), a module that increases robustness of existing diffusion based inverse problem solvers to both increase the performance and the robustness against choice of hyperparameters. DiffStateGrad projects the measurement gradients onto the low-rank approximation of intermediate states of the diffusion process (noisy manifolds for each timestep). Effectiveness of DiffStateGrad is demonstrated through multiple datasets (ImageNet and FFHQ), wide range of linear and non-linear inverse problems and applied to several SOTA posterior sampling algorithms (PSLD, ReSample and DAPS).

**Strengths:**

* The paper is written very well. The work is contextualized well among related work and I enjoyed reading the paper.
* DiffStateGrad is formulated as a module that greatly increases robustness of existing SOTA posterior sampling approaches (such as PSLD and DAPS) with respect to the choice of step siz and measurement noise.
* The effectiveness of DiffStateGrad is demonstrated in diverse set of forward models such as box inpainting, random inpainting, Gaussian deblur, motion deblur, etc. for linear inverse problems and phase retrieval, nonlinear deblur and HDR for nonlinear inverse problems.
* Additional cost of calculating SVD and projecting gradients seems negligible based on Figure 4.

**Weaknesses:**

* See the questions below.

**Questions:**

* In line 214, there is a brief comparison between MCG and DiffStateGrad, stating that one of them enforces iterates to stay close near $\mathcal{M}_t$ and the other one enforces closeness to $\mathcal{M}_0$. The fact that these methods are similar in terms of premise (measurement gradients throwing iterates off the manifold) and intuition, I think it deserves a more in depth comparison/discussion about their similarities and differences.
* How would DiffStateGrad interact with diffusion-based solvers that adopt earlier initialization strategies such as (CCDF [1] or it's adaptive version Adapt-and-Diffuse [2])? I would expect bigger step sizes are more detrimental in the "chaotic" regimes of reverse diffusion process where SNR is low. Would DiffStateGrad + CCDF combination should perform similar or better (albeit with less margin) than vanilla CCDF?
* Except for the robustness experiments, measurement noise level is not specified for the experiments (Table 2, 3, etc.). If noise was present, what was the variance and based on line 365 was it added to the images in the range [0,1] (this information is very useful for reproducibility in the future)? Does the presense/absense of measurement noise have any effect on the performance of DiffStateGrad?
  * On a related note, how does the measurement noise robustness experiment figures look when noise level is $<0.05$?

***

[1] Chung, Hyungjin, Byeongsu Sim, and Jong Chul Ye. "Come-closer-diffuse-faster: Accelerating conditional diffusion models for inverse problems through stochastic contraction." Proceedings of the IEEE/CVF Conference on Computer Vision and Pattern Recognition. 2022.

[2] Fabian, Zalan, Berk Tinaz, and Mahdi Soltanolkotabi. "Adapt and Diffuse: Sample-adaptive Reconstruction via Latent Diffusion Models." Proceedings of machine learning research 235 (2024): 12723.

---

> ### Author Response · Authors · 2024-11-21
> **Authors' Initial Response (1)**
>
> We sincerely thank the reviewer for their thoughtful and positive feedback. We kindly refer the reviewer to our general comment addressed to all reviewers, where we summarize the new experiments we conducted and highlight the key points of our work. Below, we provide a detailed response to the reviewer's specific concerns, addressing issues related to comparisons with other projection-based methods, initialization strategies, and robustness to measurement noise.
>
> **1. [Comparison to MCG]**
>
> We thank the reviewer for their comment. Below, we discuss the similarities and differences between MCG and the proposed frameworks.
>
> - MCG introduces a manifold constraint that projects the measurement gradient onto the data manifold $M_0$ corresponding to $x_{0|t}$ while applying the guidance to update $x_t$. In essence, MCG uses the concept of projection to map the gradient onto the data manifold related to $x_0$ but subsequently uses this projected gradient to update $x_t$. In contrast, our work argues that when the gradient of the measurement fidelity is used to update $x_t$, it should be projected onto a manifold that represents the noisy data at time $t$, i.e., ${\mathcal{M}}_t$. This ensures that the updates respect the state of the diffusion process at $t$. Our method is more effective as it projects the measurement gradient—used to update $x_t$—onto the noisy diffusion state associated with $x_t$, thereby preserving the $t$-state on ${\mathcal{M}}_t$ rather than ${\mathcal{M}}_0$.
>
> Please see G2 in our general response or Tables 3 and 4 for additional experiments showing that DiffStateGrad is superior to MCG. We refer the reviewer to the comparison between MCG, DPS [1], and DiffStateGrad-DPS to evaluate the impact of our proposed projection (see DPS [1] for a detailed discussion on how DPS is a generalization of the MCG method). Results show that DiffStateGrad is superior to MCG.
>
>
> **2. [DiffStateGrad when added to approaches using initialization strategies]**
>
> We thank the reviewer for highlighting the relevant literature on improving performance through better initialization strategies. We agree that such approaches can enhance the performance of solving inverse problems; however, we deliberately chose not to include these steps—such as "choosing a good initial point with the help of the measurement data"—in our experiments. This decision was made to ensure a fair comparison between each method and its corresponding DiffStateGrad without introducing additional variables. We now cite these references as an avenue to further improve DiffStateGrad for solving inverse problems.
>
> **3. [Robustness to measurement noise]**
>
> We clarify that the experiments in Tables 3-4 are conducted when there is a small measurement noise of 0.05 std. This noise level is chosen to follow a similar experiment setup to DPS [1], PSLD [2], and DAPS [3]. This is to show the impact of DiffStateGrad projection when there is little noise and the Signal-to-Noise ratio (SNR) is high. Moreover, our experiments in Figure 6 show the impact of measurement noise on the performance and the performance in general in low SNR settings. The figure shows that DiffStateGrad results in less performance decline (blue solid line) than no projection (red dashed line). We have now updated Figure 6 to include an additional point on the x-axis where the measurement noise is 0. The figure, which we show its results in numbers below, suggests that as the measurement noise level increases, the performance degradation of methods can be ameliorated by adding DiffStateGrad.
>
>
> | Method                        | SSIM (noise std = 0) ↑ | SSIM (noise std = 0.05)  ↑ |  SSIM (noise std = 0.1) ↑ | SSIM (noise std = 0.15) | SSIM (noise std = 0.2) |
> |-------------------------------|---------------|---------------|---------------|---------------|---------------|
> | PSLD                          |  **0.623**  |  **0.583**  |  0.541  | 0.516 | 0.491 |
> | DiffStateGrad-PLSD (ours)     | 0.616 | 0.580 | **0.571** | **0.553** | **0.547** |
>
>
> | Method                        | SSIM (noise std = 0) ↑ | SSIM (noise std = 0.05)  ↑ |  SSIM (noise std = 0.1) ↑ | SSIM (noise std = 0.15) | SSIM (noise std = 0.2) |
> |-------------------------------|---------------|---------------|---------------|---------------|---------------|
> | DAPS                          |  0.836 | 0.805 | 0.728 | 0.560 | 0.436 |
> | DiffStateGrad-DAPS (ours)     | **0.853** | **0.821** | **0.787** | **0.745** | **0.705** |

---

> ### Author Response · Authors · 2024-11-21
> **Authors' Initial Response (2)**
>
> We hope that our response has adequately addressed the reviewer's concerns. We look forward to hearing from the reviewer on whether their concerns have been resolved or if there are additional ways in which this paper can be further improved.
>
> ---
> [1] Chung, H.d , Kim, J., Mccann, M. T., Klasky, M. L., & Ye, J. C. Diffusion Posterior Sampling for General Noisy Inverse Problems. In The Eleventh International Conference on Learning Representations.
>
> [2] Rout, L., Raoof, N., Daras, G., Caramanis, C., Dimakis, A., & Shakkottai, S. (2024). Solving linear inverse problems provably via posterior sampling with latent diffusion models. Advances in Neural Information Processing Systems, 36.
>
> [3] Zhang, B., Chu, W., Berner, J., Meng, C., Anandkumar, A., & Song, Y. (2024). Improving diffusion inverse problem solving with decoupled noise annealing. arXiv preprint arXiv:2407.01521.

---

> > ### Comment · Reviewer_D85G · 2024-11-24
> >
> > I would like to thank the authors for their responses and for running additional experiments. I have read other reviews and responses to them. I think this is a good paper and I will maintain my score.

---

### Official Review · Reviewer_yaPA · 2024-11-04

**Soundness:** 3
**Presentation:** 3
**Contribution:** 3
**Rating:** 8
**Confidence:** 3

**Summary:**

This paper present a novel method for adding an add-on of projected gradient during solving inverse problems with (latent) diffusion models. The method can be applied for an arbitrary inverse problem solver. Results show improvement over baselines such as PSLD and ReSample for solving inverse problems with LDMs.

**Strengths:**

1. Well written
2. Easy to Understand
3. The idea of projecting the gradient to the manifold of intermediate noise is novel and making sense to me. This method supposes to suppress artifacts that arises with hard optimization.

**Weaknesses:**

1. In some cases where gradient computation may incur some additional burdens (for example when PSLD takes a lot of memory), this method may not be feasible.
2. There are some other works that try to project the restoration gradient onto the prior manifold (for example DreamClean [1], and MCG [2]).
3. The baselines with LDMs are sufficient in my view, but this paper could benefits more with pixel diffusion baselines such as DDNM [3], DDRM and so on.


[1] Xiao, Jie, et al. "DreamClean: Restoring Clean Image Using Deep Diffusion Prior." The Twelfth International Conference on Learning Representations.

[2] Chung, Hyungjin, et al. "Improving diffusion models for inverse problems using manifold constraints." Advances in Neural Information Processing Systems 35 (2022): 25683-25696

[3] Wang, Yinhuai, Jiwen Yu, and Jian Zhang. "Zero-Shot Image Restoration Using Denoising Diffusion Null-Space Model." The Eleventh International Conference on Learning Representations.

**Questions:**

1. Could authors provide more baselines results especially for pixel diffusion
2. I am curious about whether the authors consider choosing a good initial noise. Like [4, 5]


[4] Chung, Hyungjin, Byeongsu Sim, and Jong Chul Ye. "Come-closer-diffuse-faster: Accelerating conditional diffusion models for inverse problems through stochastic contraction." Proceedings of the IEEE/CVF Conference on Computer Vision and Pattern Recognition. 2022.

[5] Fabian, Zalan, Berk Tinaz, and Mahdi Soltanolkotabi. "Adapt and Diffuse: Sample-adaptive Reconstruction via Latent Diffusion Models." Proceedings of machine learning research 235 (2024): 12723.

---

> ### Author Response · Authors · 2024-11-21
> **Authors' Initial Response (1)**
>
> We sincerely thank the reviewer for their thoughtful feedback. We kindly refer the reviewer to our general comment addressed to all reviewers, where we summarize the new experiments we conducted and highlight the key points of our work. Below, we provide a detailed response to the reviewer's specific concerns, addressing issues related to computational memory requirements for gradient computation, comparisons with other projection-based methods, additional pixel-based methods, and the adoption of strategies for selecting a good initial diffusion state or noise.
>
> **1. [Computational memory]**
>
> DiffStateGrad targets a class of diffusion models that leverage measurement gradient guidance to solve inverse problems. These methods, such as DPS [1], PSLD [2], and DAPS [3], were state-of-the-art at the time of their release, and DiffStateGrad is directly applicable to these gradient-based guidance approaches. We recognize that in certain applications, the memory required for backpropagating gradients in PSLD or DPS might limit their applicability. In such cases, our method can be applied to DAPS [3], a diffusion-based approach where the measurement gradient is derived based on $x_{0|t}$ rather than $x_t$, offering a more computationally efficient alternative. In this setting, our results demonstrate that DiffStateGrad-DAPS consistently outperforms DAPS. This is evident in our initial comparisons for FFHQ (Table 3) as well as in the new comparisons on ImageNet (Table 4), where DiffStateGrad-DAPS exhibits superior performance. For the set of new results, please see our Tables in our general response (G1) above.
>
> Finally, we note that our implementation allows DiffStateGrad to be applied less frequently during diffusion iterations. For instance, in ReSample, DiffStateGrad is applied every five iterations. To further illustrate the impact of this choice, we conducted an additional ablation study, varying the projection frequency of DiffStateGrad-DAPS for random inpainting tasks on FFHQ. The results which we included in the general response (G3) indicate that reducing the frequency of projection still leads to performance improvements, demonstrating the flexibility and efficiency of our approach.
>
> **2. [Comparison to other projection-based diffusion models]**
>
> To emphasize the subspace choice, recommended by the reviewer, we have included additional comparison results with MCG [4] and MPGD [5] (see G1 in our general response). Below, we re-state our general response. The above-mentioned methods differ as follows in their respective choices of projection:
>
> - MPGD [4] employs a manifold-preserving approach to enhance the efficiency of diffusion generation and solve inverse problems. While this method aligns with a similar motivation as our proposed framework, it is specifically designed for cases where the measurement gradient is applied to $x_{0|t}$ or $z_{0|t}$. In this scenario, MPGD suggests projecting the gradient onto the tangent space of the clean data manifold. Its most effective implementation, MPGD-AE, utilizes an autoencoder (AE) for this manifold projection. However, this approach is limited by the need for an autoencoder, and its performance heavily depends on the expressive power of the autoencoder used. In contrast, DiffStateGrad is more versatile, as it can be applied to methods that use guidance on $x_{0|t}$/$z_{0|t}$ (e.g., ReSample [7] and DAPS [6]) as well as methods that apply the guidance to $x_{t}$/$z_{t}$ (e.g., PSLD [8] and DPS [1]). Notably, MPGD does not support the latter scenario. Moreover, our approach constructs the projection using SVD to capture the data structure and does not need a trained AE. We have clarified this distinction in the revised manuscript and included additional explanations in the Related Works section. We conclude that our results from new Tables 3 and 4 confirm that among pixel-based methods using the guidance to update $x_{0|t}$, DiffStateGrad-DAPS outperforms DAPS and MPGD-AE.
>
> - MCG [5] introduces a manifold constraint that projects the measurement gradient onto the data manifold $M_0$ corresponding to $x_{0|t}$ while applying the guidance to update $x_t$. In essence, MCG uses the concept of projection to map the gradient onto the data manifold related to $x_0$ but subsequently uses this projected gradient to update $x_t$. In contrast, our work argues that when the gradient of the measurement fidelity is used to update $x_t$, it should be projected onto a manifold that represents the noisy data at time $t$, i.e., ${\mathcal{M}}_t$. This ensures that the updates respect the state of the diffusion process at $t$. Our method is more effective as it projects the measurement gradient—used to update $x_t$—onto the noisy diffusion state associated with $x_t$, thereby preserving the $t$-state on ${\mathcal{M}}_t$ rather than ${\mathcal{M}}_0$.
>
> Please see G2 or Tables 3 and 4 for additional experiments showing that DiffStateGrad is superior to MCG and MPGD-AE.

---

> > ### Author Response · Authors · 2024-11-21
> > **Authors' Initial Response (2)**
> >
> > **3. [Pixel-based comparisons]**
> >
> > In response to the reviewer's suggestion, we have conducted extensive additional experiments on both the FFHQ and ImageNet datasets for pixel-based diffusion models. These new comparisons now include results from DPS, DiffStateGrad-DPS, DDNM, DDRM, MCG, and MPGD-AE for FFHQ, as well as DAPS, DiffStateGrad-DAPS, DPS, DiffStateGrad-DPS, and MPGD-AE for ImageNet. The results clearly demonstrate that DiffStateGrad outperforms all previously reported methods. We kindly refer the reviewer to our general response (G1), which includes comprehensive tables summarizing these results.
> >
> > **4. [Good initial noise]**
> >
> > We thank the reviewer for highlighting the relevant literature on improving performance through better initialization strategies. We agree that such approaches can enhance the performance of solving inverse problems; however, we deliberately chose not to include these steps—such as "choosing a good initial point with the help of the measurement data"—in our experiments. This decision was made to ensure a fair comparison between each method and its corresponding DiffStateGrad without introducing additional variables.
> >
> > We acknowledge, however, that these initialization strategies can be effectively combined with DiffStateGrad to further improve performance. To reflect this, we have now cited these works in the paper to acknowledge their potential relevance and applicability.
> >
> > To conclude, we hope that our response has adequately addressed the reviewer's concerns. We look forward to hearing from the reviewer on whether their concerns have been resolved or if there are additional ways in which this paper can be further improved. We remain open to further feedback.
> >
> > ----
> > [1] Chung, H., Kim, J., Mccann, M. T., Klasky, M. L., & Ye, J. C. Diffusion Posterior Sampling for General Noisy Inverse Problems. In The Eleventh International Conference on Learning Representations.
> >
> > [2] Rout, L., Raoof, N., Daras, G., Caramanis, C., Dimakis, A., & Shakkottai, S. (2024). Solving linear inverse problems provably via posterior sampling with latent diffusion models. Advances in Neural Information Processing Systems, 36.
> >
> > [3] Zhang, B., Chu, W., Berner, J., Meng, C., Anandkumar, A., & Song, Y. (2024). Improving diffusion inverse problem solving with decoupled noise annealing. arXiv preprint arXiv:2407.01521.
> >
> > [4] He, Y., Murata, N., Lai, C. H., Takida, Y., Uesaka, T., Kim, D., ... & Ermon, S. Manifold Preserving Guided Diffusion. In The Twelfth International Conference on Learning Representations.
> >
> > [5] Chung, H., Sim, B., Ryu, D., & Ye, J. C. (2022). Improving diffusion models for inverse problems using manifold constraints. Advances in Neural Information Processing Systems, 35, 25683-25696.
> >
> > [6] Zhang, B., Chu, W., Berner, J., Meng, C., Anandkumar, A., & Song, Y. (2024). Improving diffusion inverse problem solving with decoupled noise annealing. arXiv preprint arXiv:2407.01521.
> >
> > [7] Song, B., Kwon, S. M., Zhang, Z., Hu, X., Qu, Q., & Shen, L. Solving Inverse Problems with Latent Diffusion Models via Hard Data Consistency.
> >
> > [8] Rout, L., Raoof, N., Daras, G., Caramanis, C., Dimakis, A., & Shakkottai, S. (2024). Solving linear inverse problems provably via posterior sampling with latent diffusion models. Advances in Neural Information Processing Systems, 36.

---

> ### Comment · Reviewer_yaPA · 2024-11-22
>
> The authors address my questions well, the comparison with pixel diffusion models (e.g. + DPS/+DAPS) demonstrates that this method can generalize to both LDM and pixelDM. I am looking forward to seeing whether this method can be combined with inversion techniques / other types of models such as flow-based models. I will raise my score.

---

> > ### Author Response · Authors · 2024-11-23
> > **Authors' Response for additional MRI experiments**
> >
> > We would like to thank the reviewer for their thoughtful comment and for raising their scores. We have conducted additional experiments to show the applicability of DiffStateGrad beyond natural images and other flow-based models used for solving inverse problems. We show preliminary results on Compressed Sensing Generative Models for MRI. This is a framework which uses $x_t$ as opposed to $x_{0|t}$ for both measurement gradient computation and incorporation.
> >
> > We refer the reviewer to our new comment titled "**Additional experiment on MRI inverse problem: showing applicability beyond natural images when Langevin-based model is used**".

---

### Author Response · Authors · 2024-11-21
**Authors' Summary of General Response and Modifications to the Manuscript (1)**

We sincerely thank all the reviewers for their thoughtful and constructive feedback. We have carefully addressed each reviewer’s comments individually and incorporated their suggestions into the revised manuscript. Below, we provide a summary of the key modifications and additional analyses introduced in the paper. We believe these enhancements have strengthened our work and adequately addressed the reviewers' concerns. Finally, our changes to the manuscript can be found in color.

**G1. [Pixel-based comparison]**

In response to the reviewers' valuable suggestions, we have conducted additional experiments with "pixel-based" diffusion models and included the results in the revised manuscript. These experiments demonstrate that DiffStateGrad is not only applicable but also scalable to both latent- and pixel-based diffusion models. The results confirm that DiffStateGrad consistently enhances performance and outperforms prior work in the literature when applied to both pixel-based and latent-based models that use a guidance term to solve inverse problems. While we highlight the new pixel-based comparisons here, we kindly refer the reviewers to the manuscript for details on our initial experiments with latent diffusion models.

We have combined the previous Tables 2, 3, and 4 into a new Table 3, which now includes results for all linear and nonlinear inverse problems. In addition, we conducted new experiments for DPS [1], DiffStateGrad-DPS, and MPGD-AE [2], and reported results for DDNM [3], DDRM [4], and MCG [5] on linear inverse problems using the FFHQ dataset.

Furthermore, we included new pixel-based quantitative results in Table 4a, showcasing comparisons among DAPS [6], DiffStateGrad-DAPS, DPS, DiffStateGrad-DPS, and MPGD-AE [2] for box inpainting and random inpainting on ImageNet. Lastly, Table 4b presents additional results for DAPS, DiffStateGrad-DAPS, and MCG on Gaussian deblurring, motion deblurring, and super-resolution inverse problems conducted on ImageNet.

Below, we provide a summary of the results.

**New Results in Table 3: Performance comparison of linear tasks on FFHQ 256 x 256.**

| Method                        | LPIPS ↓ (Box-In) | PSNR ↑ (Box-In) | LPIPS ↓ (Random-In) | PSNR ↑ (Random-In) | LPIPS ↓ (Gaussian-Db) | PSNR ↑ (Gaussian-Db) | LPIPS ↓ (Motion-Db) | PSNR ↑ (Motion-Db) | LPIPS ↓ (SR) | PSNR ↑ (SR) |
|-------------------------------|---------------|--------------|------------------|-----------------|--------------------|-------------------|------------------|-----------------|--------------|-------------|
| **Pixel-based**               |               |              |                  |                 |                    |                   |                  |                 |              |             |
| DAPS                          | 0.136         | 24.57        | 0.130            | 30.79           | 0.216              | 27.92             | 0.154            | 30.13           | 0.197        | 28.64       |
| DiffStateGrad-DAPS (ours)     | **0.113**     | **24.78**    | **0.099**        | **32.04**       | 0.180          | **29.02**         | 0.119        | **31.74**       | 0.181    | **29.35**   |
| DPS                           | 0.127         | 23.91        | 0.130            | 28.67           | 0.145              | 25.48             | 0.132            | 26.75           | 0.191        | 24.38       |
| DiffStateGrad-DPS (ours)      | 0.114     | 24.10    | 0.107        | 30.15      | **0.128**          | 26.29         | **0.118**        | 27.61      | 0.186    | 24.65   |
| DDNM                          | 0.235         | 24.47        | 0.121            | 29.91           | 0.216              | 28.20             | -                | -               | 0.197        | 28.03       |
| DDRM                          | 0.159         | 22.37        | 0.218            | 25.75           | 0.236              | 23.36             | -                | -               | 0.210        | 27.65       |
| MCG                           | 0.309         | 19.97        | 0.286            | 21.57           | 0.340              | 6.72              | 0.702            | 6.72            | 0.520        | 20.05       |
| MPGD-AE                       | 0.138         | 21.59        | 0.172            | 25.22           | 0.150              | 24.42             | 0.120            | 25.72           | **0.168**    | 24.01   |

---

> ### Author Response · Authors · 2024-11-21
> **Authors' Summary of General Response and Modifications to the Manuscript (2)**
>
> **New Results in Table 4: Performance comparison of linear tasks on ImageNet 256 x 256.**
>
> **Inpainting tasks.**
>
> | Method                        | LPIPS ↓ (Box-In)      | SSIM ↑ (Box-In)       | PSNR ↑ (Box-In)       | LPIPS ↓ (Random-In)   | SSIM ↑ (Random-In)   | PSNR ↑ (Random-In)   |
> |-------------------------------|--------------------|--------------------|--------------------|--------------------|-------------------|-------------------|
> | DAPS                          | 0.217 (0.043)      | 0.762 (0.041)      | 20.90 (3.69)       | 0.158 (0.039)      | 0.794 (0.067)     | 28.34 (3.65)      |
> | DiffStateGrad-DAPS (ours)     | **0.191** (0.044)  | **0.801** (0.056)  | **21.07** (3.77)   | **0.107** (0.037)  | **0.856** (0.067) | **29.78** (4.17)  |
> | DPS                           | 0.257 (0.086)      | 0.718 (0.097)      | 19.85 (3.54)       | 0.256 (0.133)      | 0.728 (0.143)     | 26.25 (4.15)      |
> | DiffStateGrad-DPS (ours)      | 0.243 (0.093)  | 0.731 (0.100)  | 19.87 (3.61)   | 0.233 (0.138)  | 0.754 (0.150) | 27.28 (4.88)  |
> | MPGD-AE                       | 0.295 (0.057)      | 0.621 (0.053)      | 16.12 (2.26)       | 0.554 (0.148)      | 0.388 (0.112)     | 17.91 (3.25)      |
>
> **Deblurring and super-resolution tasks.**
>
> | Method                        | LPIPS ↓ (Gaussian-Db) | PSNR ↑ (Gaussian-Db)   | LPIPS ↓ (Motion-Db)   | PSNR ↑ (Motion-Db)   | LPIPS ↓ (SR)      | PSNR ↑ (SR)      |
> |-------------------------------|--------------------|---------------------|--------------------|-------------------|-------------------|------------------|
> | DAPS                          | 0.266      | 25.27        | 0.166      | 28.85     | 0.259   | 25.67     |
> | DiffStateGrad-DAPS (ours)     | **0.243**  | **25.87**    | **0.143** | **29.71**  | **0.229**  | **26.40**  |
> | MCG                           | 0.550         | 16.32           | 0.758           | 5.89          | 0.637         | 13.39        |
>
>
> **New Results in Table 6: SSIM comparison of linear tasks on FFHQ 256 x 256.**
>
> | Method                        | SSIM ↑ (Box-In)       | SSIM ↑ (Random-In)    | SSIM ↑ (Gaussian-Db)  | SSIM ↑ (Motion-Db)    | SSIM ↑ (SR)       |
> |-------------------------------|--------------------|--------------------|--------------------|--------------------|-------------------|
> | DAPS                          | 0.806 (0.028)      | 0.829 (0.022)      | 0.786 (0.051)      | 0.837 (0.040)      | 0.797 (0.044)     |
> | DiffStateGrad-DAPS (ours)     | **0.849** (0.029)  | **0.887** (0.023)  | **0.803** (0.044)  | **0.853** (0.028)  | **0.801** (0.039) |
> | DPS                           | 0.810 (0.039)      | 0.815 (0.045)      | 0.709 (0.062)      | 0.754 (0.056)      | 0.675 (0.071)     |
> | DiffStateGrad-DPS (ours)      | 0.831 (0.039)  | 0.852 (0.046)  | 0.739 (0.062)  |  0.782 (0.056)  | 0.683 (0.073) |
> | MPGD-AE                       | 0.753 (0.029)      | 0.731 (0.050)      | 0.664 (0.071)      | 0.723 (0.061)      | 0.670 (0.070)     |
>
> **G2. [How our method differs from other projection method]**
>
> Our initial Table 1 provides a brief study on the impact of subspace selection. The results indicate that the improvement in performance is not solely due to the low-dimensional nature of the gradient but rather the specific choice of projection. The table demonstrates that projecting onto the state of the diffusion process, to which the gradient guidance is applied, significantly enhances performance compared to random projections or low-rank approximations of the gradient.
>
> To further highlight the importance of subspace selection, we have included additional comparative results with MCG [5] and MPGD [2] (see above tables). These methods differ as follows in their respective choices of projection:

---

> ### Author Response · Authors · 2024-11-21
> **Authors' Summary of General Response and Modifications to the Manuscript (3)**
>
> - MPGD [2] employs a manifold-preserving approach to enhance the efficiency of diffusion generation and solve inverse problems. While this method aligns with a similar motivation as our proposed framework, it is specifically designed for cases where the measurement gradient is applied to $x_{0|t}$ or $z_{0|t}$. In this scenario, MPGD suggests projecting the gradient onto the tangent space of the clean data manifold. Its most effective implementation, MPGD-AE, utilizes an autoencoder (AE) for this manifold projection. However, this approach is limited by the need for an autoencoder, and its performance heavily depends on the expressive power of the autoencoder used. In contrast, DiffStateGrad is more versatile, as it can be applied to methods that use guidance on $x_{0|t}$/$z_{0|t}$ (e.g., ReSample [7] and DAPS [6]) as well as methods that apply the guidance to $x_{t}$/$z_{t}$ (e.g., PSLD [8] and DPS [1]). Notably, MPGD does not support the latter scenario. Moreover, our approach constructs the projection using SVD to capture the data structure and does not need a trained AE. We have clarified this distinction in the revised manuscript and included additional explanations in the Related Works section. We conclude that our results from new Tables 3 and 4 confirm that among pixel-based methods using the guidance to update $x_{0|t}$, DiffStateGrad-DAPS outperforms DAPS and MPGD-AE.
>
> - MCG [5] introduces a manifold constraint that projects the measurement gradient onto the data manifold $M_0$ corresponding to $x_{0|t}$ while applying the guidance to update $x_t$. In essence, MCG uses the concept of projection to map the gradient onto the data manifold related to $x_0$ but subsequently uses this projected gradient to update $x_t$. In contrast, our work argues that when the gradient of the measurement fidelity is used to update $x_t$, it should be projected onto a manifold that represents the noisy data at time $t$, i.e., ${\mathcal{M}}_t$. This ensures that the updates respect the state of the diffusion process at $t$. Thus, our method is more effective as it projects the measurement gradient—used to update $x_t$—onto the noisy diffusion state associated with $x_t$, thereby preserving the $t$-state on ${\mathcal{M}}_t$ rather than ${\mathcal{M}}_0$ (See G5 and the new Table 2).
>
> We also refer readers to DPS [1], which discusses how DPS generalizes MCG. We finally conclude that DiffStateGrad-DPS consistently outperforms both DPS and MCG, as evidenced by the new results presented in Table 3.
>
> **New Results in Table 3: Performance comparison of linear tasks on FFHQ 256 x 256.**
>
> | Method                        | LPIPS ↓ (Box-In) | PSNR ↑ (Box-In) | LPIPS ↓ (Random-In) | PSNR ↑ (Random-In) | LPIPS ↓ (Gaussian-Db) | PSNR ↑ (Gaussian-Db) | LPIPS ↓ (Motion-Db) | PSNR ↑ (Motion-Db) | LPIPS ↓ (SR) | PSNR ↑ (SR) |
> |-------------------------------|---------------|--------------|------------------|-----------------|--------------------|-------------------|------------------|-----------------|--------------|-------------|
> | **Pixel-based**               |               |              |                  |                 |                    |                   |                  |                 |              |             |
> | DAPS                          | 0.136         | _24.57_        | 0.130            | 30.79           | 0.216              | 27.92             | 0.154            | 30.13           | 0.197        | 28.64       |
> | DiffStateGrad-DAPS (ours)     | **0.113**     | **24.78**    | **0.099**        | **32.04**       | 0.180          | **29.02**         | 0.119        | **31.74**       | 0.181    | **29.35**   |
> | DPS                           | 0.127         | 23.91        | 0.130            | 28.67           | 0.145              | 25.48             | 0.132            | 26.75           | 0.191        | 24.38       |
> | DiffStateGrad-DPS (ours)      | 0.114     | 24.10    | 0.107        | 30.15      | **0.128**          | 26.29         | **0.118**        | 27.61      | 0.186    | 24.65   |
> | MCG                           | 0.309         | 19.97        | 0.286            | 21.57           | 0.340              | 6.72              | 0.702            | 6.72            | 0.520        | 20.05       |
> | MPGD-AE                       | 0.138         | 21.59        | 0.172            | 25.22           | 0.150              | 24.42             | 0.120            | 25.72           | **0.168**    | 24.01   |

---

> ### Author Response · Authors · 2024-11-21
> **Authors' Summary of General Response and Modifications to the Manuscript (4)**
>
> **G3. [Efficiency of the proposed method]**
>
> In response to concerns regarding the efficiency of our projection method as image size increases, we would like to highlight the following points. First, our pixel-based analysis demonstrates that DiffStateGrad applies to both pixel-based diffusion models (e.g., DPS and DAPS) and latent-based diffusion models (e.g., PSLD and ReSample). Additionally, our implementation includes a projection period parameter, which allows projections to be performed less frequently. For instance, while our experiments with DAPS apply projections at every iteration, the ReSample experiments use projections every five iterations. This information has been included in the Appendix for reference.
>
> We also conducted a new ablation study on DiffStateGrad-DAPS for random inpainting tasks on FFHQ where we find that increasing the projection period (decreasing the frequency) still leads to performance improvement. We include these results in the new Figure 10 in the Appendix. The results, summarized below, demonstrate the flexibility and efficiency of our approach.
>
> | Method                        | LPIPS ↓ | SSIM  ↑ |  PSNR ↑ |
> |-------------------------------|---------------|---------------|---------------|
> | DAPS                          |  0.130   |  0.829  |  30.79  |
> | DiffStateGrad-DAPS (ours) P=1    | **0.099**  |  **0.887**  |   **32.04** |
> | DiffStateGrad-DAPS (ours) P=5   | 0.116   | 0.863   |    31.41 |
> | DiffStateGrad-DAPS (ours) P=10   |  0.118  |  0.860   |  31.36 |
>
> where P denotes the projection period.
>
> **G4. [Increasing Robustness of Diffusion-based Inverse Solvers]**
>
> Finally, we would like to re-emphasize one of the key practical benefits of DiffStateGrad: its ability to increase the robustness of diffusion-based models to the choice of gradient step size as well as measurement noise. While the literature continues to propose new and more sophisticated approaches, their complexity and the extensive hyperparameter tuning they require pose significant challenges for experimentalists and scientists looking to adopt these frameworks in their fields. In contrast, DiffStateGrad represents a practical step towards making diffusion models more accessible and user-friendly.
>
> Figure 5 illustrates the increased robustness to measurement gradient step size in PSLD introduced by DiffStateGrad. Furthermore, Figure 6a highlights the improved robustness of DAPS when DiffStateGrad is applied, particularly against measurement noise levels for box inpainting. Similarly, Figure 6b demonstrates the enhanced robustness of PSLD against measurement noise levels for Gaussian deblurring when using DiffStateGrad. Given one reviewer's suggestion, Figure 6 is now modified to include results when the measurement noise is 0. The figure, which we show its results in numbers below, suggests that as the measurement noise level increases, the performance degradation of methods can be ameliorated by adding DiffStateGrad.
>
>
> | Method                        | SSIM (noise std = 0) ↑ | SSIM (noise std = 0.05)  ↑ |  SSIM (noise std = 0.1) ↑ | SSIM (noise std = 0.15) | SSIM (noise std = 0.2) |
> |-------------------------------|---------------|---------------|---------------|---------------|---------------|
> | PSLD                          |  **0.623**  |  **0.583**  |  0.541  | 0.516 | 0.491 |
> | DiffStateGrad-PLSD (ours)     | 0.616 | 0.580 | **0.571** | **0.553** | **0.547** |
>
>
> | Method                        | SSIM (noise std = 0) ↑ | SSIM (noise std = 0.05)  ↑ |  SSIM (noise std = 0.1) ↑ | SSIM (noise std = 0.15) | SSIM (noise std = 0.2) |
> |-------------------------------|---------------|---------------|---------------|---------------|---------------|
> | DAPS                          |  0.836 | 0.805 | 0.728 | 0.560 | 0.436 |
> | DiffStateGrad-DAPS (ours)     | **0.853** | **0.821** | **0.787** | **0.745** | **0.705** |
>
> **G5. [Prior works guidance computation and guidance incorporation]**
>
> To provide further clarity on how our method relates to prior works, we have added an additional paragraph to the Background and Related Works section. This paragraph categorizes prior works based on three key criteria: (a) how the measurement guidance gradient is computed (GC), (b) how the gradient guidance is incorporated into the diffusion process (GI), and (c) whether the method utilizes a projection.
>
> For convenience, we restate this new paragraph in the next comment, along with the updated Table 2.

---

> ### Author Response · Authors · 2024-11-21
> **Authors' Summary of General Response and Modifications to the Manuscript (5)**
>
> "Prior works differ from one another in two key aspects: (a) how the gradient is computed and (b) how the gradient is used to update the diffusion state. Table 2 categorizes prior works based on these characteristics. While a few approaches, such as diffusion-based MRI, compute the measurement gradient using $x_{t}$, most recent literature has shifted toward using $x_{0|t}$ for gradient computation. Regarding gradient incorporation, the literature is further subdivided. For instance, methods like DPS and PSLD use the measurement gradient to update the state at time $t$, whereas other approaches, such as ReSample, DAPS, and MPGD, apply the guidance to the conditional state at $0|t$ before resampling. Additionally, while the projections in MCG and MPGD are restricted to $x_{0|t}$, DiffStateGrad stands out by being applicable to both types of methods."
>
> **New Table 2**
> | Method                 | GC  ($x_{0 given t}$ or $z_{0 given t}$)   | GI ($x_t$ or $z_t$)    |  GI  ($x_{0 given t}$ or $z_{0 given t}$)  | Proj   ($x_t$ or $z_t$)   |  Proj  ($x_{0 given t}$ or $z_{0 given t}$)    |
> |-------------------------------|-----------------------|-----------------------|----------------------|----------------------|-----------------------|
> | DPS                 | ✓   | ✓   | ✗  | ✗ | ✗ |
> | PSLD                   | ✓    | ✓   | ✗  | ✗ | ✗ |
> | ReSample               | ✓    | ✗   | ✓  | ✗ | ✗ |
> | DAPS                   | ✓    | ✗   | ✓  | ✗ | ✗ |
> | MCG                     | ✓    | ✓   | ✗  | ✗ | ✓ |
> | MPGD                     | ✓    | ✗   | ✓  | ✗ | ✓ |
> | **DiffStateGrad (ours)** | ✓ | ✓   | ✓  | ✓ | ✓ |
>
> [1] Chung, H., Kim, J., Mccann, M. T., Klasky, M. L., & Ye, J. C. Diffusion Posterior Sampling for General Noisy Inverse Problems. In The Eleventh International Conference on Learning Representations.
>
> [2] He, Y., Murata, N., Lai, C. H., Takida, Y., Uesaka, T., Kim, D., ... & Ermon, S. Manifold Preserving Guided Diffusion. In The Twelfth International Conference on Learning Representations.
>
> [3] Wang, Y., Yu, J., & Zhang, J. Zero-Shot Image Restoration Using Denoising Diffusion Null-Space Model. In The Eleventh International Conference on Learning Representations.
>
> [4] Kawar, B., Elad, M., Ermon, S., & Song, J. (2022). Denoising diffusion restoration models. Advances in Neural Information Processing Systems, 35, 23593-23606.
>
> [5] Chung, H., Sim, B., Ryu, D., & Ye, J. C. (2022). Improving diffusion models for inverse problems using manifold constraints. Advances in Neural Information Processing Systems, 35, 25683-25696.
>
> [6] Zhang, B., Chu, W., Berner, J., Meng, C., Anandkumar, A., & Song, Y. (2024). Improving diffusion inverse problem solving with decoupled noise annealing. arXiv preprint arXiv:2407.01521.
>
> [7] Song, B., Kwon, S. M., Zhang, Z., Hu, X., Qu, Q., & Shen, L. Solving Inverse Problems with Latent Diffusion Models via Hard Data Consistency.
>
> [8] Rout, L., Raoof, N., Daras, G., Caramanis, C., Dimakis, A., & Shakkottai, S. (2024). Solving linear inverse problems provably via posterior sampling with latent diffusion models. Advances in Neural Information Processing Systems, 36.

---

> ### Author Response · Authors · 2024-11-21
> **Authors' Summary of General Response and Modifications to the Manuscript (6)**
>
> We thank the reviewers for their constructive feedback. We hope that the above discussion, along with our detailed responses to the reviewers' feedback, have adequately addressed their concerns. We look forward to hearing from the reviewers on whether their concerns have been resolved or if there are additional ways in which this paper can be further improved.

---

### Author Response · Authors · 2024-11-21
**Authors' Summary of Manuscript Modifications**

We thank all reviewers for their constructive feedback. Given our extensive new analyses, we have made several changes to the manuscript.  We refer the reviewers to the changes in color green in the manuscript, and summarize below the key changes

- New Table 2: We have created a new Table 2 that categorizes prior works based on gradient guidance computation, incorporation, and projection. This table highlights that DiffStateGrad is applicable to both $x_t$/$z_t$ and $x_{0|t}$/$z_{0|t}$.

- Revised Table 3: Tables 2, 3, and 4 have been merged into a single Table 3. Table 3a now includes both pixel-based (6 new rows) and latent-based diffusion models for all linear tasks on FFHQ. Table 3b reports our previous results for nonlinear tasks on FFHQ. This table now focuses on LPIPS and PSNR metrics, with SSIM values moved to the Appendix (see Table 6).

- New Table 4a: This table contains results for both pixel-based (5 new rows) and latent-based diffusion models for inpainting tasks on ImageNet.

- New Table 4b: This table includes results for pixel-based (3 new rows) and latent-based diffusion models for deblurring and super-resolution tasks on ImageNet.

- Figure Reorganization: To accommodate the new results, we have moved the original Figure 7 (“Performance of DiffStateGrad-DAPS for different variance retention thresholds”) to the Appendix, where it is now labeled as Figure 9.

 - Expanded Related Works: The Related Works section has been updated with additional information on MCG and MPGD, clarifying how our proposed framework differs from these methods.

- Reproducibility: We have included additional details on the new experiments in the Appendix to support reproducibility.

- Additional Citations: We have added citations for methods suggested by reviewers: [1-2]: Approaches that accelerate or improve diffusion-based inverse solvers through initialization strategies. [3]: A method focused on inpainting inverse problems using a gradient-free masking approach to generate only missing pixels.

- Visualization Adjustments: To save space for additional experimental results, half of the visualizations from Figure 7 have been moved to the Appendix.

- Figure 10: We have included an additional figure for the numbers we report on the impact of the projection period (See G3).

We hope the reviewers find this summary helpful, and we refer the reviewer to the manuscript for more details.

[1] Chung, H., Sim, B., & Ye, J. C. (2022). Come-closer-diffuse-faster: Accelerating conditional diffusion models for inverse problems through stochastic contraction. In Proceedings of the IEEE/CVF Conference on Computer Vision and Pattern Recognition (pp. 12413-12422).

[2] Fabian, Z., Tinaz, B., & Soltanolkotabi, M. (2024). Adapt and Diffuse: Sample-adaptive Reconstruction via Latent Diffusion Models. Proceedings of machine learning research, 235, 12723.

[3] Lugmayr, A., Danelljan, M., Romero, A., Yu, F., Timofte, R., & Van Gool, L. (2022). Repaint: Inpainting using denoising diffusion probabilistic models. In Proceedings of the IEEE/CVF conference on computer vision and pattern recognition (pp. 11461-11471).

---

### Author Response · Authors · 2024-11-23
**Additional experiment on MRI inverse problem: showing applicability beyond natural images when Langevin-based model is used**

We would like to thank the reviewer yaPA for their thoughtful comment and for raising their scores. As suggested by reviewer yaPA, we have conducted additional experiments. This is to demonstrate the applicability of DiffStateGrad beyond natural image tasks and to showcase its compatibility with other guidance-based Langevin dynamics used for solving inverse problems.

We conducted additional experiments on Compressed-Sensing Magnetic Resonance Imaging (MRI). This represents an inverse problem where undersampled measurements are observed in the frequency domain, and the task is to reconstruct high-quality MR images in the image space.

For these experiments, we utilized the Compressed Sensing Generative Model (CSGM) proposed in [3], which employs Langevin dynamics for MRI reconstruction (available at https://github.com/utcsilab/csgm-mri-langevin). We incorporated our proposed DiffStateGrad into the measurement gradient guidance. Given the complex nature of MRI data, we applied DiffStateGrad to the magnitude of the complex-valued data while preserving the phase information. This framework uses $x_t$ for both measurement gradient computation and incorporation. The unconditional diffusion model was trained on T2-weighted brain datasets from the NYU fastMRI dataset [1-2], and the reported results were averaged over 30 test examples (reporting avg (std)). The measurements were sampled vertically and equispaced at an undersampling rate of $R=4$.

While the default optimized measurement gradient step size from [3] is $5.0$, our results confirm that DiffStateGrad enhances the robustness of CSGM-Langevin when the step size deviates from this optimal value. For example, CSGM-Langevin's performance is significantly reduced when measurement gradient step size increases (see step size = 10), while DiffStateGrad's performance decline is less. This suggests that DiffStateGrad can be a practical tool to reduce the risk of drastic failure in real-world applications.

We will update the manuscript shortly to include these new results. Given the small sample size and the preliminary nature of this analysis, we will include these results in the Appendix, provide a brief reference to them in the main text and leave extensive analysis for future work.


**Step size = 2.5**
| Method                        | PSNR ↑ | SSIM ↑ | NMSE  ↓ |  LPIPS ↓ |
|-------------------------------|---------------|--------------|------------------|------------------|
CSGM-Langevin | 34.85 ( 4.4739 ) | 0.8345 ( 0.1308 ) | 0.03072 ( 0.0709 ) | 0.1474 ( 0.0618 )|
DiffStateGrad-CSGM-Langevin (Ours) | 34.78 ( 4.3892 ) | 0.8423 ( 0.1277 )| 0.02881 ( 0.0637 ) | 0.153 ( 0.0745 ) |


**Step size = 5.0** (the default value chosen by CSMRI-Langevin)
| Method                        | PSNR ↑ | SSIM ↑ | NMSE  ↓ |  LPIPS ↓ |
|-------------------------------|---------------|--------------|------------------|------------------|
CSGM-Langevin | 34.31 ( 3.5739 ) | 0.796 ( 0.1119 ) | 0.02922 ( 0.0701 ) | 0.1797 ( 0.0546 ) |
DiffStateGrad-CSGM-Langevin (Ours) | 33.75 ( 3.9638 ) | 0.8021 ( 0.1256 ) | 0.03241 ( 0.0643 ) | 0.1716 ( 0.0611 )|


**Step size = 7.5**
| Method                        | PSNR ↑ | SSIM ↑ | NMSE  ↓ |  LPIPS ↓ |
|-------------------------------|---------------|--------------|------------------|------------------|
CSGM-Langevin | 32.88 ( 2.9101 ) | 0.723 ( 0.1023 ) | 0.03324 ( 0.0723 ) | 0.2465 ( 0.0593 )|
DiffStateGrad-CSGM-Langevin (Ours) | 33.21 ( 3.3072 ) | 0.7768 ( 0.1156 ) | 0.03043 ( 0.053 ) | 0.1882 ( 0.051 ) |


**Step size = 10.0**
| Method                        | PSNR ↑ | SSIM ↑ | NMSE  ↓ |  LPIPS ↓ |
|-------------------------------|---------------|--------------|------------------|------------------|
CSGM-Langevin | 31.38 ( 2.5123 ) | 0.6552 ( 0.0948 ) | 0.04107 ( 0.0769 ) | 0.3014 ( 0.0599 ) |
DiffStateGrad-CSGM-Langevin (Ours) | 32.49 ( 3.0165 ) | 0.7386 ( 0.1136 ) | 0.03181 ( 0.0478 ) | 0.2111 ( 0.0491 )|


[1] Knoll, F., Zbontar, J., Sriram, A., Muckley, M. J., Bruno, M., Defazio, A., ... & Lui, Y. W. (2020). fastMRI: A publicly available raw k-space and DICOM dataset of knee images for accelerated MR image reconstruction using machine learning. Radiology: Artificial Intelligence, 2(1), e190007.

[2] Zbontar, J., Knoll, F., Sriram, A., Murrell, T., Huang, Z., Muckley, M. J., ... & Lui, Y. W. (2018). fastMRI: An open dataset and benchmarks for accelerated MRI. arXiv preprint arXiv:1811.08839.

[3] Jalal, A., Arvinte, M., Daras, G., Price, E., Dimakis, A. G., & Tamir, J. (2021). Robust compressed sensing mri with deep generative priors. Advances in Neural Information Processing Systems, 34, 14938-14954.

---

> ### Author Response · Authors · 2024-11-26
> **Revised manuscript uploaded**
>
> We have now incorporated the above into the manuscript. Please see Appendix E.

---

### Meta-Review · Area_Chair_JJTj · 2024-12-14

**Metareview:**

The starting point of the paper is very reasonable. Specifically, the authors aim to relieve the issue that previous methods fail to preserve the generation process on the data manifold defined by the diffusion prior when solving the inverse problem. The authors empirically find that the crucial factor is to choose the low-rank subspace of the latent image representation for containing the gradient, which is meaningful. Three of the reviewers provided very positive feedback, while one has doubts about the paper’s contributions.
The AC tends to agree with the opinions of the first three reviewers that the overall quality of this paper is good. However, after discussing with the reviewers, we believe that the assumption of Proposition 1 may be too strong, which may be misleading, and the claimed theoretical contribution to be reasonable. Thus, I only recommend the paper to be accepted as a  poster paper.

**Additional Comments On Reviewer Discussion:**

The authors provide Proposition 1, which aims to offer theoretical support that DiffStateGrad helps the samples remain on or close to the manifold. However, the theoretical result is based on a crucial assumption: “the projection operator onto an approximate subspace closely approximates the projection onto the tangent space.” This assumption seems almost equivalent to directly assuming that the theoretical result is true, which makes the theoretical result appear to lack significance and necessity.

---

### Decision · Program_Chairs · 2025-01-22

Accept (Poster)